# Evaluation of a new snow albedo scheme for the Greenland ice sheet in the regional climate model RACMO2

Christiaan T. van Dalum[1], Willem Jan van de Berg[1], Stef Lhermitte[2], and Michiel R. van den Broeke[1]

[1]Institute for Marine and Atmospheric Research, Utrecht University, Utrecht, The Netherlands
[2]Department of Geoscience & Remote Sensing, Delft University of Technology, Delft, The Netherlands

**Correspondence:** Christiaan van Dalum (c.t.vandalum@uu.nl)

**Abstract.** Snow and ice albedo schemes in present day climate models often lack a sophisticated radiation penetration scheme and do not explicitly include spectral albedo variations. In this study, we evaluate a new snow albedo scheme in the regional climate model RACMO2 for the Greenland ice sheet, version 2.3p3, that includes these processes. The new albedo scheme uses the two-stream radiative transfer in snow model TARTES and the spectral-to-narrowband albedo module SNOWBAL, version 1.2. Additionally, the bare ice albedo parameterization has been updated. The snow and ice broadband and narrowband albedo output of the updated version of RACMO2 is evaluated using PROMICE and K-transect in-situ data and MODIS remote-sensing observations. Generally, the modeled narrowband and broadband albedo is in very good agreement with satellite observations, leading to a negligible domain-averaged broadband albedo bias for the interior. Some discrepancies are, however, observed close to the ice margin. Compared to the previous model version, RACMO2.3p2, the broadband albedo is considerably higher in the bare ice zone during the ablation season, as atmospheric conditions now alter the bare ice broadband albedo. For most other regions, however, the updated broadband albedo is lower due to spectral effects, radiation penetration or enhanced snow metamorphism.

## 1 Introduction

The absorption of shortwave radiation is an important component of the surface energy balance of snow-covered surfaces (Van den Broeke et al., 2005; He et al., 2018b; Warren, 2019). A drop in the surface reflectivity for solar radiation, i.e., albedo, leads to more absorbed energy in the snowpack, which in turn leads to higher snow temperatures or melt. This melt-albedo feedback is initiated once snow starts to melt and the snow structure is altered, lowering the albedo (Van As et al., 2013; Jakobs et al., 2019). It is therefore imperative for regional and global climate models (RCMs and GCMs, respectively) to capture snow albedo correctly in order to reproduce the current climate and to make future climate projections for snow covered glaciated regions such as the Greenland ice sheet (GrIS).

The albedo of snow and ice is highly spectrally dependent and also depends on various other quantities. For clean snow, the spectral albedo, i.e. the albedo as a function of wavelength, is almost one for near-ultraviolet (near-UV, 300-400 nm) and visible light (400-750 nm), but drops for near-infrared (near-IR, 750-1400 nm) and is low and fluctuating for infrared (IR) radiation (Fig. 1d, Warren and Wiscombe (1980); Warren et al. (2006); Gardner and Sharp (2010); Dang et al. (2015); Picard et al.

(2016)). Impurities like soot and dust lower the spectral albedo significantly in the near-UV and visible part of the spectrum (Hansen and Nazarenko, 2004; Doherty et al., 2010; Dumont et al., 2014; Tuzet et al., 2017). Snow metamorphism, which leads to increased snow density and grain radius, alters the albedo as well, especially for the (near-)IR radiation (Wyser and Yang, 1998; King et al., 2004; Tuzet et al., 2019; He and Flanner, 2020). With coarser grains, light has to travel longer through ice before it has the opportunity to reflect off a grain's surface out of the snowpack than for fine-grained snow, hence lowering the albedo (Wiscombe and Warren, 1980; Gardner and Sharp, 2010; Picard et al., 2012; Warren, 2019). Fresh snow with a small grain radius, for example, has a high albedo (typically larger than 0.8), while firn and ice, for which the grain radius has grown due to metamorphism, have a lower albedo (typically approximately 0.55 for ice and 0.7 for firn). Likewise, snow grain shape impacts the probability for light to reflect out of the snowpack (Libois et al., 2013; He et al., 2018a), but Dang et al. (2016) show that a model with spherical grains can still accurately reproduce the measured spectral albedo by adjusting the grain radius. To summarize, it is thus essential to consider the spectral albedo of snow and ice when modeling the snowpack or ice melt.

Since incoming solar radiation also varies greatly as a function of wavelength (Gates, 1966; Leckner, 1978), the broadband albedo, i.e., the wavelength-integrated spectral albedo, is also altered by atmospheric properties, like clouds and water vapour, and by the solar zenith angle (SZA) (Dang et al., 2015). The SZA impacts the spectral distribution of incoming light, as Rayleigh scattering by the atmosphere is more effective for shorter wavelengths, but also alters the angle of incidence into the snowpack (Solomon et al., 1987; Gardner and Sharp, 2010; Van Dalum et al., 2019). A large SZA results in a shallow angle of incidence, increasing the probability for light to scatter out of the snowpack, which increases the spectral albedo. In addition, upper snow layers are often characterized by small grains, enhancing the spectral albedo even further. However, this increase of spectral albedo at large SZA is partly mitigated by the red shift of the incoming direct-beam radiation due to enhanced Rayleigh scattering in the atmosphere. During cloudy conditions, radiation is more likely to scatter, changing the weighted average SZA and thus the spectral albedo. Furthermore, clouds and water vapour alter the spectral distribution of radiation at the surface by filtering out IR radiation. Subsequently, the blue shift of incoming radiation under cloudy conditions leads to an increase of the broadband albedo even if the snow structure remains unaltered.

RCMs and GCMs commonly perform their radiative calculations for the atmosphere on a limited number of spectral bands. The albedo of such a spectral band is defined as the narrowband albedo. Some of these models do conduct coarsely-resolved spectral calculations on a few bands, like E3SM and CESM (Caldwell et al., 2019; Danabasoglu et al., 2020), but more often they do not use narrowband albedos and determine a broadband albedo instead, bypassing its spectral bands and neglecting any spectral albedo variations. Recently, progress has been made in the development of new snow albedo parameterzations and coupling schemes, which allows for the use of spectral bands and more physical processes to be included (Libois et al., 2013; Van Dalum et al., 2019).

In this study, we improve the snow and ice albedo parameterization in the polar version of the Regional Atmospheric Climate Model (RACMO2) and present version 2.3p3. The polar (p) version of RACMO2 is a model developed to simulate the climate and atmosphere-surface interaction of glaciated regions, in particular Greenland (e.g., Noël et al., 2018) and Antarctica (e.g., Van Wessem et al., 2018). The snow albedo scheme of previous RACMO2 versions (2.1 to 2.3p2) used an adjusted version

of the parameterization of Gardner and Sharp (2010) to derive a broadband albedo. Therefore, RACMO2 until now did not include explicit spectral albedo or spectral irradiance effects, nor an adequate radiation penetration scheme. Introducing a new snow albedo parameterization that includes these processes is therefore timely.

RACMO2.3p3 uses a new snow albedo parameterization using the Two-streAm Radiative TransfEr in Snow model (TARTES, Libois et al., 2013) coupled with the Spectral-to-NarrOWBand ALbedo (SNOWBAL) module version 1.2 (Van Dalum et al., 2019). This set-up provides appropriate narrowband albedos for all 14 shortwave spectral bands utilized in RACMO2. TARTES also considers radiation penetration for its surface albedo calculations and provides estimates of energy absorption in the snowpack. Additionally, the new snow albedo parameterization is used to develop a new ice albedo scheme.

Here, we present and evaluate the broadband and narrowband albedo modeled by RACMO2.3p3 for the GrIS, and compare it with remote sensing data, in-situ observations and the broadband albedo modeled by the previous iteration of RACMO2, version 2.3p2. The remainder of this manuscript is made up of six sections. Section 2 summarizes the changes made in RACMO2 and introduces the remote sensing and in-situ observational data sets. Section 3 and Sect. 4 evaluate the new RACMO2 version with these data sets. Comparisons between the albedo modeled by RACMO2 version 2.3p3 and 2.3p2 are shown in Sect. 5. Finally, the sensitivity to the chosen impurity concentration of snow is analyzed in Sect. 6, and results are discussed and conclusions are drawn in Sect. 7. The impact of the model improvements on the climate, surface mass balance and surface energy balance of the GrIS ice sheet will be discussed in a forthcoming publication.

## 2  Model and observational data sets

### 2.1  Regional climate model

The Regional Atmospheric Climate Model (RACMO2), integrates the atmospheric dynamics of the High Resolution Limited Area Model (HIRLAM, Undén et al., 2002), version 5.0.3, with the surface and atmospheric processes of the European Center for Medium-Range Weather Forecasts (ECMWF) Integrated Forecast System (IFS), cycle 33r1 (ECMWF, 2009). The polar version of RACMO2, version 2.3p2, from now on abbreviated to Rp2, is adapted for glaciated tiles by using a multilayer snowpack that interacts with the atmosphere and involves processes within the snow column, such as melt and refreezing. Rp2 is introduced in more detail in Noël et al. (2018). At the lateral boundaries, RACMO2 is forced with ERA-Interim data (Dee et al., 2011). In the new RACMO2 version, 2.3p3, from now on abbreviated to Rp3, two components have been adjusted, the multilayer firn module and the snow and ice albedo parameterizations for glaciated regions.

### 2.1.1  Multilayer firn module updates

In Rp3, the multilayer firn module has been rewritten to improve code efficiency and reduce numerical diffusion. As the surface albedo depends on the structure of the snowpack, any changes made to the multilayer firn module are therefore also important to discuss. The update of this module consists of four modifications.

Firstly, Rp2 used a prognostic fresh snow layer, which is effectively a sublayer of the uppermost snow model layer. In Rp3 this fresh snow layer is removed; instead, the uppermost layers are allowed to be very thin, i.e., in the order of millimeters, thus containing fresh snow only. For heat diffusion calculations, these thin layers are treated as a single layer to maintain numerical stability. If melt or refreezing occurs in one of these layers, their individual temperature is estimated obeying the temperature gradient and conserving their combined heat content.

Secondly, in Rp2 layers below a threshold thickness merged with the first layer below. In Rp3, a layer merges with the adjacent layer having most similar density and grain size. Furthermore, undesired numerical diffusion is avoided by implementing mass redistribution if a thin layer merges with a thick layer. A layer containing glacial ice is not allowed to merge with layers that are formed locally, i.e., by snow deposition on this grid point. This allows for the formation and preservation of layers with ice lenses.

Thirdly, internal energy absorption heats subsurface snow layers and can induce melt. In Rp3, melt will only thin a subsurface snow layer, i.e., a layer with a density below $700 \, \mathrm{kg \, m^{-3}}$, and not change its density. For ice layers, i.e., with a layer density larger than $830 \, \mathrm{kg \, m^{-3}}$, melt creates pore space, reducing the layer density and no thinning occurs. For firn with intermediate densities, the induced layer thinning fraction linearly decreases from 1 to 0 between $700 \, \mathrm{kg \, m^{-3}}$ and $830 \, \mathrm{kg \, m^{-3}}$. The resulting density is adjusted accordingly. Melting of the uppermost layer always leads to thinning, regardless of its density.

Finally, the initialized ice density is increased from $910 \, \mathrm{kg \, m^{-3}}$ to $917 \, \mathrm{kg \, m^{-3}}$, which is more in agreement with observations (Bader, 1964), and is used to convert the effective grain radius into a SSA. Furthermore, as ice layers melt, pore space is created, which lowers the layer density. The lower density for bare ice layers then indicates that air bubbles are present within the ice.

### 2.1.2   Snow albedo

Rp2 used a plane-parallel broadband snow albedo scheme based on Gardner and Sharp (2010), that depended indirectly on
wavelength in the form of tuning parameters, and is limited to two snow layers. This albedo scheme parameterized albedo variations due to a changing SZA, grain radius, cloud cover, impurities, and an altitude-dependent atmospheric optical thickness, the latter for clear-sky conditions (Kuipers Munneke et al., 2011). In RACMO2, the first two snow layers are often very thin, i.e. a few millimeters for fresh snow and up to a few centimeters for older snow, thus effectively neglecting almost all radiation penetration.

In Rp3, the Two-streAm Radiative TransfEr in Snow model (TARTES, Libois et al., 2013) coupled with the Spectral-to-NarrOWBand ALbedo (SNOWBAL) module (Van Dalum et al., 2019) is implemented. TARTES is a spectral albedo model based on the radiative transfer equation (Jiménez-Aquino and Varela, 2005) and asymptotic analytical radiative transfer theory (Kokhanovsky, 2004; He and Flanner, 2020) using the geometric-optics method, which allows for a vertically inhomogeneous snowpack. Grain radius, grain shape, snow layer density, impurity concentration and type, and SZA are all explicitly resolved.
In this study, all grains are spherically shaped. TARTES is able to calculate a spectral albedo for any wavelength between 199 and 3003 nm and returns the absorption of radiation within the snowpack for both incoming direct, i.e., no atmospheric scattering, and diffuse radiation, i.e., light that is scattered by the atmosphere, of which the latter is considered to be a direct beam with a SZA of $53°$.

In order to couple TARTES with the IFS physics within RACMO2, which employs 14 contiguous shortwave spectral bands (Fig. 1d), the SNOWBAL module has been developed (Van Dalum et al., 2019). Since both the spectral albedo and the incoming solar radiation can vary within a spectral band, SNOWBAL selects the predefined representative wavelengths for the given atmospheric condition that would provide the correct effective narrowband albedos by TARTES. Using simply the wavelength of the center of the spectral bands increases the root-mean-square error (RMSE) of the broadband albedo by approximately 0.05 and 0.04 for clear-sky direct and clear-sky diffuse radiation, respectively, and increases even more for cloudy conditions (Van Dalum et al., 2019). The representative wavelength depends on the SZA for clear-sky diffuse radiation, SZA and vertically integrated water vapour for clear-sky direct radiation, and ice and liquid water path for cloudy conditions. The difference between cloudy-diffuse and clear-sky diffuse albedo are thus only related to cloud and SZA induced spectral shifts in radiation. Furthermore, direct radiation dominates the clear-sky albedo signal except for very high SZA. As full radiation calculations are only performed every hour, the average SZA of the next hour is used as long as the sun is above the horizon. Excluded are bands 13 and 14, for which the narrowband albedo can safely assumed to be zero (Gardner and Sharp, 2010; Van Dalum et al., 2019). For the other bands, three narrowband albedos are determined, i.e., for direct and diffuse radiation for clear-sky conditions, and for diffuse radiation for cloudy conditions. Clear-sky and total-sky narrowband and broadband albedos are then determined using the modeled radiative fluxes. Note that clear-sky and total-sky albedo are identical if no clouds are present. Finally, for evaluation with the seven MODIS narrowband albedos, clear-sky diffuse radiation albedos are also explicitly derived for these bands.

In this manuscript, 'albedo' without further specification refers to the broadband albedo. Clear-sky direct (CSdir) and clear-sky diffuse (CSD) albedo refers to surfaces illuminated only by direct radiation or diffuse radiation, respectively. Combined, they are referred to clear-sky albedo. The clear-sky and cloudy-sky albedo can in turn be combined to a total-sky albedo.

### 2.1.3 Bare ice albedo

In Rp2, a background bare ice albedo (BIA) field is defined for the entire ice sheet and used if one of the upper two snow layers are identified as bare ice. The BIA field is prescribed by the lowest five percent of the 16-day diffuse albedo product (MCD43A3v5, Schaaf and Wang (2015)) of 1 km MODIS data, for the period 2000-2015 (Fig. 1a). The MODIS albedo field is resampled to the model grid, and the BIA is limited to values between 0.3 for dark ice in the ablation zone, and 0.55 in the accumulation zone under perennial snow (Noël et al., 2018).

As we do not want to bypass TARTES for bare ice, we derived a representative specific surface area (SSA) and impurity concentration field to be used for bare ice albedo calculations to resemble the broadband MODIS albedos. Firstly, we assume that clean blue ice has an albedo of approximately 0.6 (Reijmer et al., 2001; Dadic et al., 2013). Blue ice is typically found in areas with a very smooth surface and high sublimation rates, but no melt, and has a high bubble content, leading to a relatively high albedo. The bare ice albedo is subsequently lowered by standing water, surface roughness and impurities. Furthermore, we assume that MODIS bare ice albedos are valid for clear-sky conditions (Wang et al., 2012; Casey et al., 2017). TARTES, however, does not use Mie-scattering theory, which would be preferable for ice (Gardner and Sharp, 2010) and cannot model bubbles or liquid water explicitly. Hence, albedos ranging from 0.30 to 0.55 observed for the GrIS are obtained by increasing

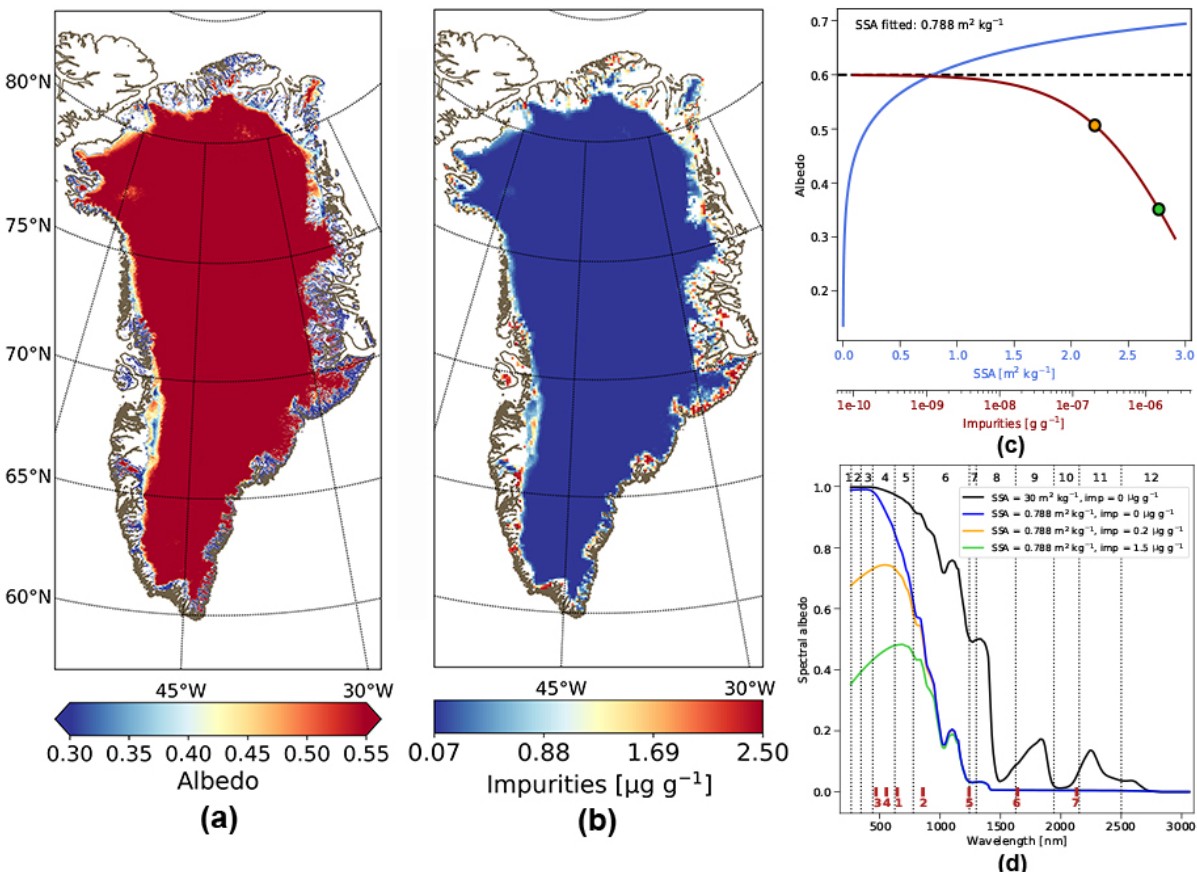

**Figure 1. (a)** Lowest five percent of the MODIS MCD43A3v5 1 km 16-day clear-sky diffuse albedo field for glaciated areas for the period 2000-2015. As this albedo field is used to determine a bare ice albedo field, it is limited between 0.30 for dark ice in the ablation zone, and 0.55 in the accumulation zone under perennial snow for consistency with RACMO2 (Noël et al., 2018). **(b)** Bare ice impurity field that is implemented in RACMO2.3p3 for glaciated grid points. Here, all impurities are soot. **(c)** In blue, fitting the specific surface area (SSA) to clean blue ice albedo, which is assumed to be 0.6. The fitted SSA equals 0.788 m² kg⁻¹. In red, the soot concentration as a function of albedo required to successfully convert the MODIS albedo field into an impurity field. For both lines, clear-sky conditions are assumed for a SZA of 60° and RACMO2 irradiance profiles are used to convert narrowband to broadband albedo. **(d)** Spectral albedo for clean fresh snow (in black), and for an ice profile with the fitted SSA of 0.788 m² kg⁻¹ for various impurity concentrations. The first twelve spectral bands of RACMO2 are indicated by vertical dotted lines and black numbers. Red bars and numbers indicate the seven MODIS spectral bands. The albedo for the cases with soot concentrations of 0.2 and 1.5 µg g⁻¹ are indicated with corresponding colored dots in **(c)**.

the soot content, with the absorption cross section for soot that is determined by Kokhanovsky (2004). Using a semi-infinite layer with the density of ice, a SSA value of 0.788 m² kg⁻¹ (4.152 mm grain size, which is an order of magnitude larger than the typical grain radius for snow (Warren, 2019)) is found to provide an albedo 0.6 (Fig. 1b). Despite not using Mie-scattering theory, the spectral curve in TARTES for this SSA value resembles the expected curve for bare ice quite well, especially in the

(infra)red part of the spectrum (Fig. 1d, blue line) (e.g., Dadic et al., 2013), and thus can be used to indicate clean, bare ice, which is similar to the findings of Bohren (1983).

Next, the MODIS bare ice albedo range is converted to an impurity concentration. Using the fitted SSA, ice density and a reference SZA of 60°, which is the largest angle for which the observations of MODIS for the GrIS are still somewhat reliable (Wang and Zender, 2010), together with RACMO2 narrowband irradiance profiles for such conditions, a broadband albedo can be calculated for a range of impurities such that the MODIS albedo range is covered. The resulting soot concentration varies between 69 ng g$^{-1}$ for an albedo of 0.55, to 2445 ng g$^{-1}$ for an albedo of 0.3 (Fig. 1c), and are saved in a lookup table. This lookup table is used to convert the MODIS albedo field, after resampling to the 11-km grid of RACMO2, to an impurity field feasible for TARTES to use in RACMO2 (Fig. 1b). Adding soot alters, as expected (Doherty et al., 2010; Gardner and Sharp, 2010; He and Flanner, 2020), only the spectral curve in the near-UV and visible part of the spectrum (Fig. 1d). Note that the broadband albedo can still reach values beyond the range indicated in Fig. 1c, depending on atmospheric conditions and SZA.

### 2.1.4 Superimposed ice

In Rp3, superimposed ice is now distinct from glacial ice. Superimposed ice forms in snow layers by refreezing of melt water, while glacial ice forms by compaction of snow. As superimposed ice has a granular structure, it has to be treated differently than bare ice. In Rp3, superimposed ice forms in snow or firn layers, where the grain radius is allowed to grow due to refreezing of melt. Due to the granular structure of superimposed ice, it is desirable to use the snow albedo scheme of Sect. 2.1.2 over the bare ice albedo scheme of Sect. 2.1.3, as a fixed rather large grain radius is used for bare ice. However, without additional corrections the typical grain size used in superimposed ice layers is 0.7 to 1.0 mm, leading to unrealistically high albedos of approximately 0.7 (Granskog et al., 2006). In order to improve this, a minimum grain radius for superimposed ice is imposed which increases linearly from 0.720 mm for a density of 750 kg m$^{-3}$, to 4.152 mm for a density of 917 kg m$^{-3}$. This correction leads to realistic albedos for superimposed ice and exposed ice lenses.

In Rp3, superimposed ice is treated differently than glacial ice. Superimposed ice forms in snow layers by refreezing of melt water, while glacial ice forms by compaction of snow. As superimposed ice has a granular structure (Granskog et al., 2006), it has to be treated differently than bare ice. Due to the granular structure, it is desirable to use the snow albedo scheme of Sect. 2.1.2 over the bare ice albedo scheme of Sect. 2.1.3, as a fixed rather large grain radius is used for bare ice. However, without additional corrections, the typical grain size of model layers with superimposed ice in Rp3 is 0.7 to 1.0 mm, leading to unrealistically high albedos of approximately 0.7. The typical albedo of superimposed ice is 0.65, as is measured by Knap and Oerlemans (1996) at S9 of the K-transect. In order to improve this, a minimum grain radius for superimposed ice is imposed, which increases linearly from 0.720 mm for a density of 750 kg m$^{-3}$, to the bare ice value of 4.152 mm for a density of 917 kg m$^{-3}$. This superimposed ice layer uses the same impurity concentration as a snow layer. This correction leads to realistic albedos for superimposed ice and exposed ice lenses.

## 2.2 RACMO2 simulations

For all simulations in this manuscript, RACMO2 is run on an 11-km grid of Greenland and its immediate surroundings, for the period 2006 - 2015, using September 2000 to 2005 as spin-up. At the lateral boundaries, RACMO2 is forced with ERA-Interim data (Dee et al., 2011). The only impurity type considered is soot, with a prescribed concentration of 5 ng g$^{-1}$ for all snow layers. Although the concentration of soot in Greenland varies considerably over time and space, RACMO2 only allows for a fixed soot concentration in snow. If a layer is identified as bare ice, it is prescribed by the spatially variable soot concentration of Fig. 1b. For snow in the interior and when no melt occurs, the soot concentration is approximately 3 ng g$^{-1}$ (McConnell et al., 2007; Doherty et al., 2010; Dang et al., 2015). The impact of impurities, however, is known to be underestimated by TARTES, so a higher prescribed concentration is required to model it properly (Tuzet et al., 2017, 2019). The impact of soot is assessed in various sensitivity experiments, which are done between 2011 and 2015, with September 2008 to 2011 as spin-up.

For initialization, the firn-column state, i.e., the layer thickness, snow and ice density, water concentration, temperature and grain size, were taken for all active layers from the Rp2 run on the initialization day, i.e., 1 September 2000, but the fresh snow sub-layer data is omitted. In order to match the specifications of Rp3, glacial ice is identified in each firn column if the continuous set of layers has a density of 899 kg m$^{-3}$ or higher, counted from the bottom of the firn column. Furthermore, Rp3 is initialized with a soot concentration that can be used to calculate the bare ice albedo (Sect. 2.1.3).

## 2.3 MODIS snow albedo product

The RACMO2 albedo product is evaluated with the Moderate Resolution Imaging Spectroradiometer (MODIS), using the MCD43A3 Version 006 Albedo Model daily dataset using 16-day Terra and Aqua MODIS data for white-sky, i.e., clear-sky diffuse (CSD), and black-sky, i.e., clear-sky direct (CSDir), conditions (Schaaf and Wang, 2015). CSD albedo and CSDir albedo are calculated for local solar noon. CSD albedo is preferred for evaluation, as no uncertainties arise concerning SZA, although the difference between the CSD and CSDir albedo product is only marginal (Williamson et al., 2016). While RACMO2 calculates the direct and diffuse albedo, it only produces total-sky and clear-sky albedo output, which includes both direct and diffuse radiation. Therefore, we have to note that the MCD43A3 CSD albedo product remains a slightly different albedo product than the clear-sky RACMO2 albedo it evaluates. MCD43A3 provides an albedo product in seven shortwave bands, ranging between 620 and 2155 nm (Fig. 1d, red bars), with a spatial resolution of 500 m (250 m for band 1 and 2). In addition, visible, near-IR and broadband albedo products are provided. In this study, the broadband albedo and seven shortwave band albedos are used for model evaluation.

For comparison with RACMO2, MODIS data are resampled to the 11-km grid of RACMO2. Due to the lack of a proper ice mask for this MODIS field, contamination with non-glaciated grid points cannot be excluded for some grid points at the margin. Therefore, these grid points are omitted if the albedo becomes too low (smaller than 0.25) during summer.

Lhermitte et al. (2014) and Williamson et al. (2016) reported that MODIS captures the albedo evolution well for most of the GrIS, but that it has problems for inhomogeneous regions, like mountain ranges. MODIS albedo also shows a drop in accuracy for SZAs larger than 55°, and becomes physically unrealistic for SZAs larger than 65° (Wang and Zender, 2010; Liu

et al., 2009). Therefore, the evaluation with RACMO2 is limited to a SZA of 55° or less. In addition, latitudes north of 75°N should be excluded, as the noise-to-signal ratio of MODIS becomes too high (Wang and Zender, 2010; Manninen et al., 2019). Consequently, northern Greenland is omitted from the evaluation. As MODIS only measures during clear-sky conditions, some
230 regions have limited coverage. Extensive evaluation of the MCD43A3 Version 006 Albedo product shows that it compares well with observations (Wright et al., 2014; Burkhart et al., 2017; Moustafa et al., 2017; Wang et al., 2018). For Summit located in central Greenland, Wright et al. (2014) report a RMSE and mean albedo difference with respect to in-situ observations of 0.026 and 0.015, respectively, indicating that MCD43 slightly overestimates the albedo.

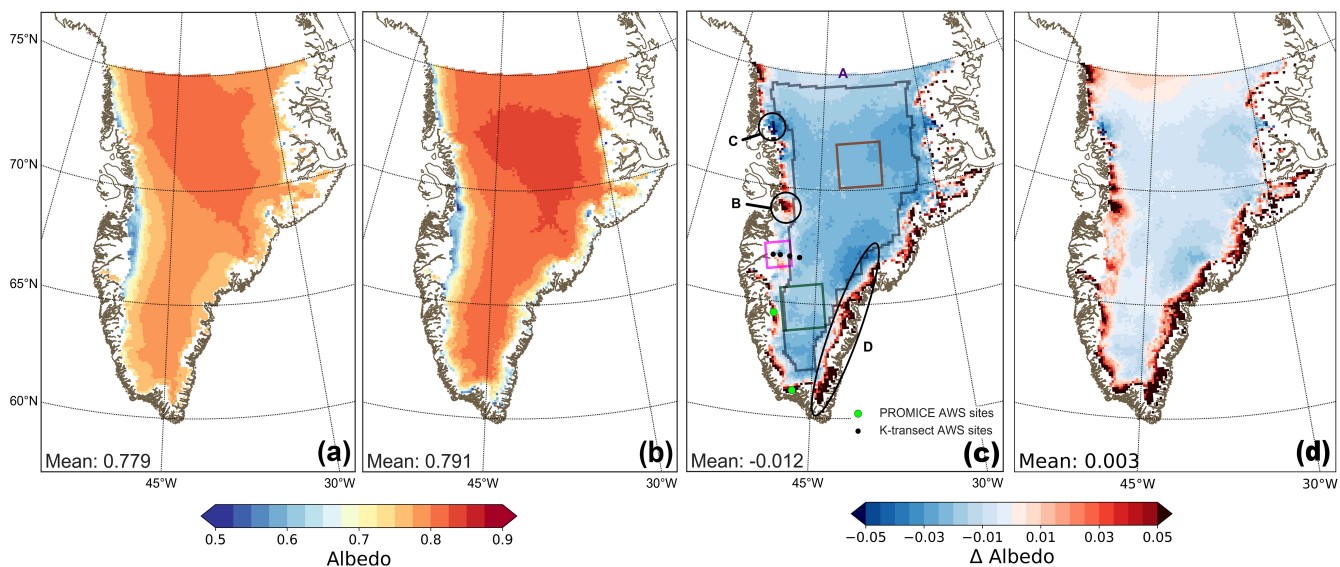

**Figure 2. (a)** Average 16-days running mean clear-sky RACMO2.3p3 albedo for 15:00 UTC (12:00 LT for most of Greenland) between 2006 and 2015, **(b)** MODIS MCD43A3 clear-sky diffuse (CSD) albedo, **(c)** albedo difference between RACMO2.3p3 clear-sky albedo and MODIS MCD43A3 CSD albedo and **(d)**, like **(c)**, but now after applying a uniform -0.015 bias correction to the MODIS MCD43A3 data. AWS locations of the K-transect are indicated in **(c)** by the black dots, from west to east: S5, S6, S9 and S10. PROMICE AWS NUK-U (north) and QAS-U (south) are indicated by the light-green dots. KAN-U and KAN-M are located close to S10 and between S6 and S9, respectively, but are not shown separately. Extended evaluation is done for the enclosed purple region that is indicated by **A**, for areas **B**, **C** and **D**, and the regions enclosed in the colored boxes.

## 2.4 In-situ measurements

In-situ observations provide insight in the performance of RACMO2 for total-sky and cloudy conditions, unavailable from remote sensing observations. Therefore, we evaluate the albedo product with the Automatic Weather Station (AWS) data along the Kangerlussuaq-transect (K-transect, Smeets et al., 2018) and with a selection of AWS data of the Programme for Monitoring of the Greenland Ice Sheet (PROMICE) (Van As et al., 2011).

The K-transect data used are from AWS sites S5, S6, S9 and S10, and are available for 2006 up to 2015, with an exception
for S10, which is only available between 2010 up to 2015. The K-transect is located in southwest Greenland (Fig. 2c) around
67° N. More specifically, S5 and S6 are located in the ablation zone, S9 approximately on the equilibrium line, and S10 in the
accumulation zone. Daily hourly-averaged observations are considered at noon local time in Greenland (15:00 UTC), and a
running average of 16 days is taken to fit the temporal sampling of MODIS.

PROMICE AWS are mostly located in the ablation zone. Only AWS sites that cover at least partially the model period and
for which an appropriate grid point can be selected in RACMO2, are selected. Figure 2c shows the location of the PROMICE
stations used in this study. KAN-U and KAN-M are located along the K-transect, NUK-U and QAS-U are located in the
southwest and south, respectively.

## 3 Evaluation using MODIS albedo

In this section, we evaluate and discuss the Rp3 clear-sky albedo output with MODIS CSD albedo for both broadband and
narrowband albedo and discuss processes involved.

### 3.1 Comparison with MODIS broadband albedo

Figure 2 shows the 2006-2015 average 16-days running mean clear-sky albedo of Rp3 (Fig. 2a) and MODIS CSD albedo
(Fig. 2b) at 15:00 UTC (local noon for most of Greenland). On average, the spatial patterns are similar, while some local
differences can be observed (Fig. 2c). The domain-averaged bias considering all glaciated grid points is -0.012, indicating a
slight underestimation of the modeled albedo with respect to MODIS CSD albedo. For the interior, indicated by **A** and enclosed
within the purple line in Fig. 2c, which excludes all grid points within five grid points of the margin, we observe an average
bias of -0.022, which is close to the mean difference of -0.015 for Summit reported by Wright et al. (2014). Correcting for
this MCD43 mean albedo difference (Fig. 2d), the bias for area **A** reduces to -0.007, supporting excellent agreement in the
accumulation zone. Furthermore, only a small overestimation is observed in the large bare ice region around the K-transect
(black dots in Fig. 2c), showing that the new bare ice albedo parameterization produces adequate results for this region.

Around Jakobshavn (circle indicated by **B** in Fig. 2c), Rp3 considerably overestimates the albedo, especially during the
accumulation season. This region is characterized by rough and heavily crevassed terrain, causing an inhomogeneous snow
cover. Rp3, however, evenly distributes snow within a grid cell, that is, patches of snow are not modeled. Consequently,
the albedo of Rp3 is too high, as snow-free areas like hummocks and crevasse openings are not properly captured. Snow
subsequently remains for too long at the surface in Rp3 before melting in the ablation season.

The region indicated by **C** shows an underestimation of Rp3 albedo compared to MODIS. Rp3 typically only models a few
decimeters of snow during winter that melt quickly during summer, thinning and eventually removing these layers. Throughout
the summer, Rp3 likely underestimates the snow thickness, leading to underestimated albedo.

Most pronounced biases can be observed for grid points close to the margin in the southeast (region **D**), but extending
beyond this region. The albedo difference is typically around 0.03, but can be as high as 0.15. The ablation zone in **D** is too

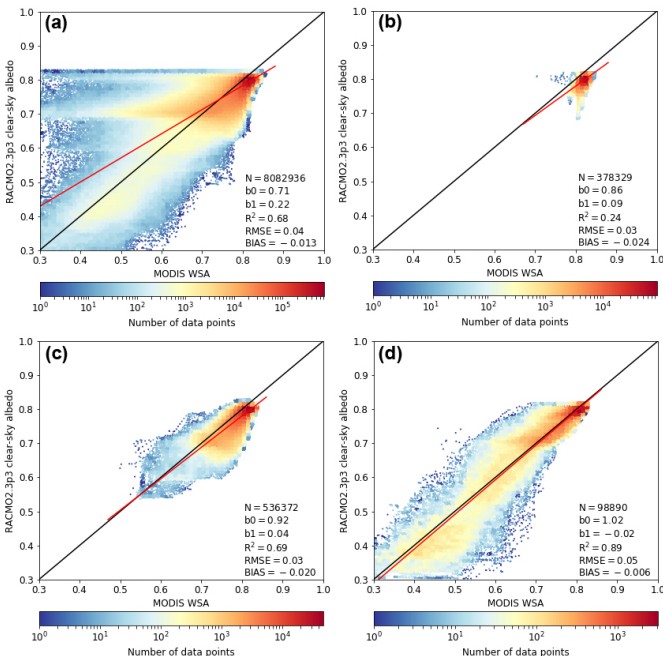

**Figure 3.** Comparison between 16-days running mean RACMO2.3p3 clear-sky albedo and MODIS MCD43A3 CSD albedo. Every data point represents an observation at a grid point at 15:00 UTC for a day between 2006 and 2015. In black the 1-to-1 line. The red line is the linear regression of the data, with b0 the slope and b1 the intercept. In addition, number of records (N), correlation coefficient ($R^2$) and root-mean-square error (RMSE) are displayed. Colours indicate number of data points and the bin size is 0.01. **(a)** Displays data for the whole domain, **(b)**, **(c)** and **(d)** the brown, green and pink boxes in Fig. 2c, respectively.

narrow (up to 10 km) to be adequately resolved at the used resolution of 11 km (Noël et al., 2015). Furthermore, the southeast is characterized by heavy snowfall (Ohmura and Reeh, 1991; Mernild et al., 2015), and in combination with the low model resolution, snow persists at the surface throughout the ablation season in Rp3. Still, the grain radius increases and the albedo drops somewhat during the ablation season, but not as fast as the albedo decline in MODIS. Note that the uncertainty of MODIS
is also considerable, as this is mountainous terrain (see Sect. 2.3).

Figure 3 correlates the uncorrected MODIS CSD albedo with Rp3 clear-sky albedo at noon on a grid-point level for the whole time span and domain (Fig. 3a) and in various regions (Fig. 3b-d). In general, Rp3 correlates well with MODIS with a low RMSE and bias, but typically underestimates the albedo, as is also observed in Fig. 2. With declining albedos, both Rp3 and MODIS CSD albedo generally follow the same pattern, also in the bare ice regime. There are, however, two distinct outliers
visible in Fig. 3a, around Rp3 albedos of 0.81 and 0.70, where Rp3 models fresh and melting snow, respectively, instead of bare ice that MODIS observes. These data points originate from the southeast (area **D** of Fig. 2c, see the previous discussion). The bias for the Summit region (Fig. 3b) is similar to both the bias of area **A** (Fig. 2c) and the bias of the MODIS CSD albedo product (Wright et al., 2014). Furthermore, the albedo variability is low for this region, as melt does normally not occur here

(except for July 2012 (Nghiem et al., 2012; Bennartz et al., 2013)) and metamorphism is slow in this cold climate. Figure 3c represents an area in southern Greenland without bare ice, which is characterized by albedo decrease due to rapid snow metamorphism (Lyapustin et al., 2009). Rp3 performs well for this region. The region shown in Fig. 3d corresponds roughly with the ablation zone of the K-transect. The bias is very low considering the large variability in the ablation zone and the correlation coefficient is high. Both the bare ice and snow albedo schemes perform well and merge smoothly together. Note that the outliers of Fig. 3a are absent around the K-transect, indicating that it is not the bare ice albedo scheme that causes the discrepancies.

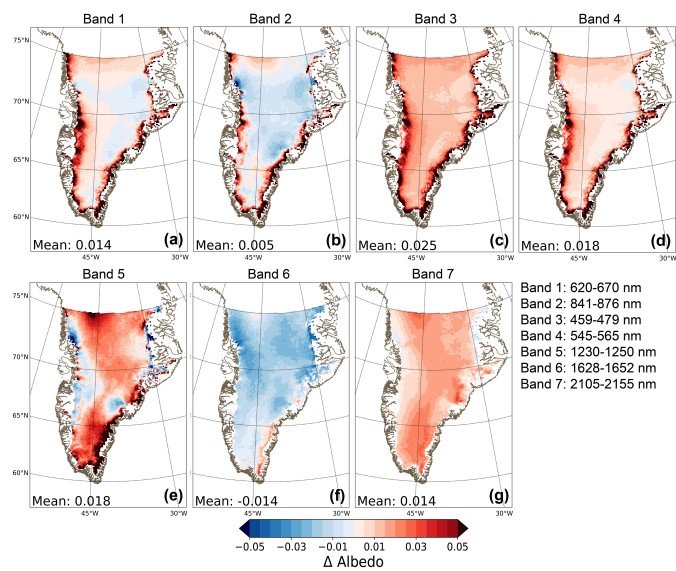

**Figure 4.** Average albedo difference for 15:00 UTC between RACMO2.3p3 16-days running mean CSD albedo and MODIS MCD43A3 CSD product, between 2006 and 2015 for **(a)** band 1 to **(g)** 7 (red bands in Fig. 1d). The spectral width of each band is also shown.

## 3.2   Comparison with MODIS narrowband albedo

MODIS broadband albedo is derived from its seven narrowband sensors (red bands in Fig. 1d). Note that the quality of band 6 is reduced for the AQUA satellite due to instrument failure (Stroeve et al., 2006; Box et al., 2012). To allow for direct comparison with MODIS' narrowband observations, TARTES within Rp3 is also run for wavelengths representative for the seven MODIS bands with diffuse radiation. For these bands, CSD albedo output of Rp3 is available. Note that the albedo determined for the seven MODIS bands are not used to compute a broadband albedo within Rp3. Figure 4 shows the mean spectral albedo difference for these seven bands. Bands that are associated with a strong albedo gradient as a function of wavelength (Fig. 1d) show larger spatial variations (band 2 and 5, Fig. 4b and e) than bands that either have a high (band 1, 3 and 4, Fig. 4a, c and d) or low albedo (band 6 and 7, Fig. 4f and g). Overall, differences in the interior are small. Large differences are again observed in area **D**. Bands 1, 3 and 4 show a large positive model bias for the bare ice zone, while the bias for this region is limited

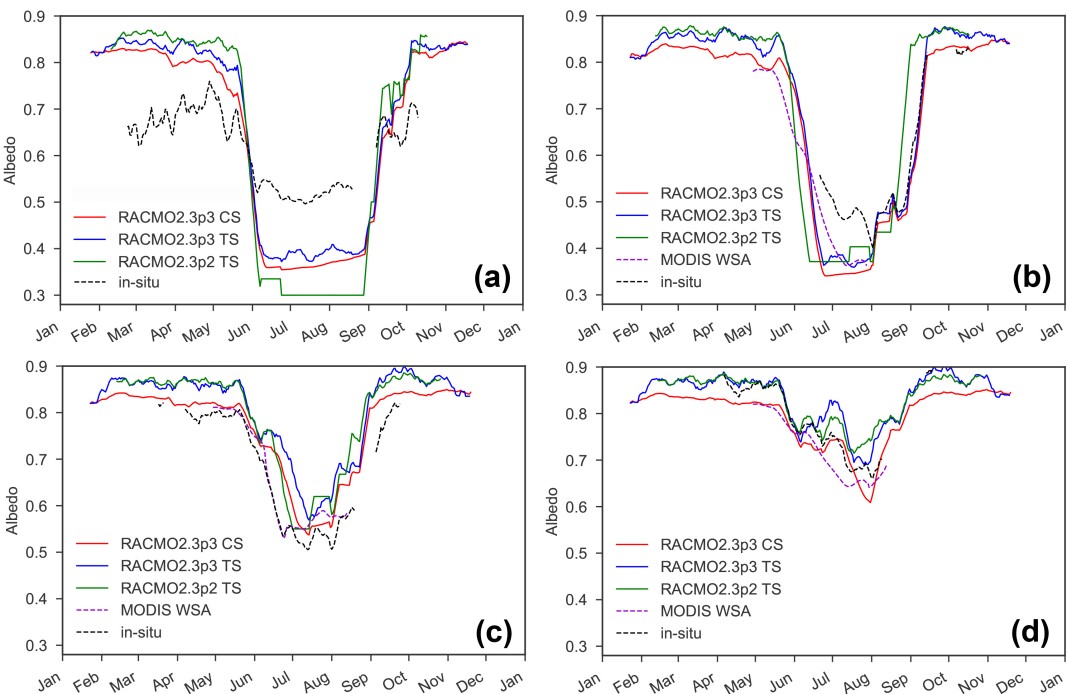

**Figure 5.** Time series of the average 16-days running mean broadband albedo for 15:00 UTC for RACMO2.3p3 clear-sky (CS) and total-sky (TS) conditions, RACMO2.3p2 TS, MODIS clear-sky diffuse (CSD) albedo and in-situ albedo for 2012 for **(a)** S5, **(b)** S6, **(c)** S9 and **(d)** S10.

for bands 2, 5, 6 and 7. This positive bias for bands 1, 3 and 4 is only present when bare ice is at the surface, illustrating the importance of correctly modeling soot in ice, as soot alters the albedo in particular for these bands (Gardner and Sharp, 2010). Although large quantities of soot have been added for bare ice, as discussed in Sect. 2.1.3, apparently it is not enough to lower the albedo for bare ice to MODIS values. It is, however, expected that Rp3 needs large amounts of soot to lower the bare ice

albedo, as no dust, cryoconite and algae are modeled, all of which lower the albedo (Bøggild et al., 2010). Elsewhere, Rp3 generally shows small differences with the seven MODIS bands, indicating that the albedo for various wavelengths is captured well.

## 4  In-situ broadband albedo measurements

Along the K-transect, albedo data are available at S5, S6, S9 and S10 (Fig. 2c) for 15:00 UTC and are shown in Fig. 5 for 2012

and compared with Rp3 clear-sky and total-sky conditions, Rp2 total-sky, and MODIS CSD. S5 is too close to the ice margin for MODIS to produce a reliable albedo at this resolution, and is not included. As the area around this station is characterized by very rough terrain, RACMO2 at this low resolution has trouble reproducing the in-situ albedo.

For S6, Rp3 albedo corresponds well with the measurements, especially when snow returns after the melt season. At the onset of the accumulation season, Rp2 performes considerably worse, as a lack of radiation penetration causes a too rapid albedo increase. As the accumulation season progresses and the snow layers become thicker, the albedo difference between Rp3 and Rp2 diminishes. MODIS CSD albedo fits well with Rp3 clear-sky albedo for bare ice conditions. The resolution of RACMO2 is not sufficient to capture all spatial variations, i.e., the AWS data might not be representative for the full grid point of RACMO2 in which S6 is located.

For S9 (Fig. 5c), both Rp3 and Rp2 total-sky albedo show relatively large deviations with the in-situ measurements during the accumulation season, but the difference of Rp3 clear-sky albedo with MODIS CSD albedo is much smaller. Site S9 is characterized by spatial inhomogeneity, although to a lesser degree than S5 and S6. During the start of the ablation season, superimposed ice persists at the surface, delaying the albedo drop to bare ice values by a few days. Just like at S6, the Rp3 albedo fits better than Rp2 with both in-situ and MODIS observations at the onset of snowfall after summer.

S10 observations (Fig. 5d), which are representative for the lower accumulation zone of Greenland, correspond well with both Rp3 and Rp2. In addition, Rp3 fits with MODIS CSD albedo before the melt season, and performs reasonably well during and after the melt season. 2012 is characterized by a long and intense melt season, explaining the albedo decrease observed in the MODIS CSD albedo product. The melt season albedo decrease in Rp3, however, is slightly delayed.

## 5   Comparison with RACMO2.3p2

In this section, we compare the Rp3 albedo product with Rp2 and highlight the differences. Moreover, we investigate the impact that clouds have on the albedo and investigate the seasonal differences.

### 5.1   Broadband albedo differences

Figure 6 shows the 16-days running mean total-sky albedo for 15:00 UTC between 2006 and 2015 for Rp2 and the albedo difference between Rp3 and Rp2. For most of the ice sheet, a small negative difference is observed, i.e., the albedo of Rp3 is slightly lower than Rp2.

In areas **E** in the north-west and **F** in the north-east (Fig. 6b), the albedo of Rp3 is lower than Rp2. Both regions are characterized by limited snowfall, resulting in a shallow snowpack on top of bare ice during the accumulation season for several months (Fig. 7). At the end of the ablation season when new snow layers start to form, melt can still occur, forming melt water within those layers. This is more common in Rp3, as internal heating increases the subsurface temperature and therefore more melt occurs, changing the snow structure and lowering the albedo in Rp3 with respect to Rp2. Furthermore, if melt is strong enough, it can remove or thin the fresh snow top layer, exposing bare ice underneath. As the accumulation season progresses and the sun rises again above the horizon, the snowpack is already thick enough to prevent solar radiation to reach subsurface bare ice layers in any significant amount, reducing the albedo difference.

We also observe a small albedo difference (smaller than 0.01) in the high accumulation region in southeast Greenland (area **D** of figure 2c). For some grid points around the margins, the albedo of Rp3 is considerably higher than Rp2. This difference

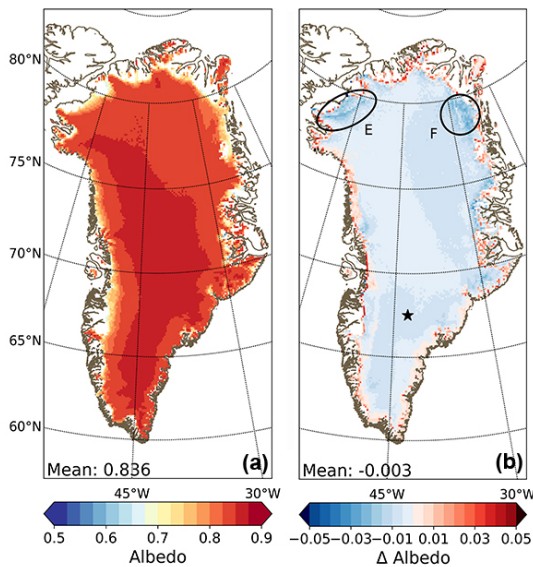

**Figure 6. (a)** Average 16-days running mean total-sky RACMO2.3p2 albedo and **(b)** the average albedo difference between RACMO2.3p3 and RACMO2.3p2, for 15:00 UTC between 2006 and 2015, with positive differences indicating that the albedo of RACMO2.3p3 is larger than RACMO2.3p2. Extended evaluation is done for the enclosed region of **E** and **F** and the grid point located at the star.

is more pronounced in areas with exposed bare ice during the ablation season, and is limited to a single grid point bordering non-glaciated tiles. These differences can be traced back to uncertainties in the bare ice albedo field of Rp2, where grid points close to the margin were contaminated with tundra albedo, and an albedo difference with the new model is therefore expected.

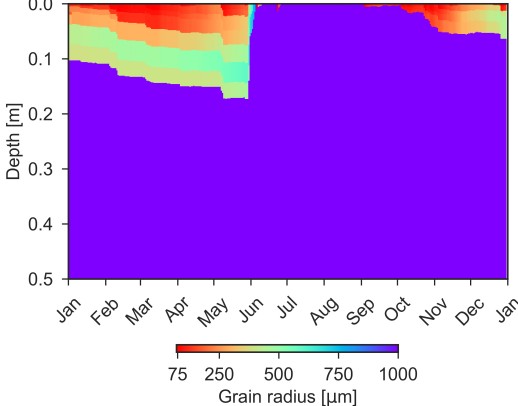

**Figure 7.** RACMO2.3p3 grain radius of the upper 20 snow layers as a function of depth for 2012, for a grid point within area **E** (Fig. 6b). Bare ice is indicated by a grain radius of 1000 μm.

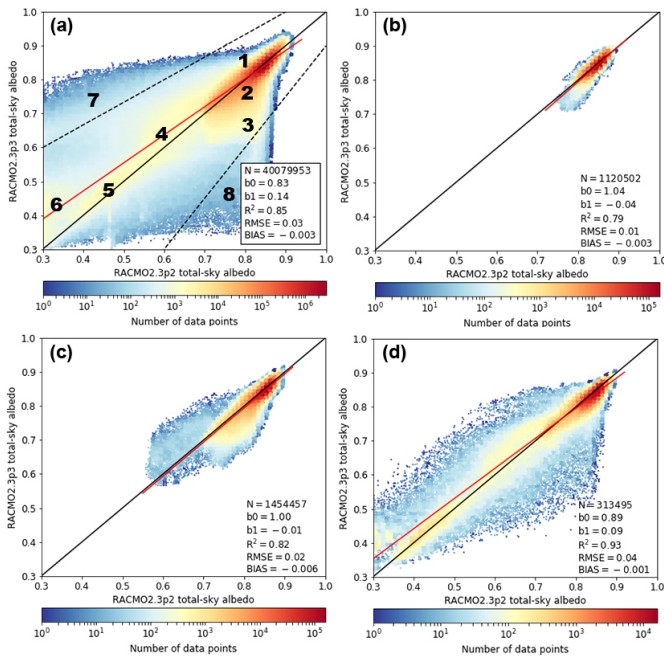

**Figure 8.** Comparison of 16-days running mean total-sky albedo of RACMO2.3p3 with RACMO2.3p2 for 15:00 UTC, between 2006 and 2015. Similar to Fig. 3, **(a)** shows all ice sheet points, **(b)**, **(c)** and **(d)** only grid cells in the brown, green and pink boxes in Fig. 2c, respectively. Numbers and dashed lines in **(a)** are discussed in the text.

Figure 8 compares albedo of Rp3 and Rp2 for the entire ice sheet and smaller regions indicated by the colored boxes in Fig. 2c. In general, Rp3 albedo correlates well with the albedo product of Rp2 (Fig. 8a), as the bulk of occurrences are close to the

1-to-1 line (black line). Most of the GrIS is covered in snow, for which the albedo is high (larger than 0.75) and the albedo difference is small (number **1** in Fig. 8a). Snow metamorphism is slightly stronger in Rp3, leading to lower albedos (about 0.75) compared to Rp2 (about 0.8, **2**). Both **1** and **2** typically occur in the interior (Fig. 8b).

Larger albedo differences occur when firn or ice are close to the surface as radiation penetration lowers Rp3 albedo (**3**). A snow profile similar to Fig. 7 in early June would result in such an albedo difference. For the 0.6 albedo bin of Rp2 (**4**), the

albedo of Rp3 is generally 0.05 higher, illustrating that Rp3 seems to have a slower firn-ice transition. In south Greenland (Fig. 8c), processes **1**, **2**, **3** and **4** are all relevant.

The snow and ice albedo merge well, but the Rp3 albedo is often higher than Rp2 for the ablation zone, i.e., for albedos smaller than 0.6. Still, highest occurrence density is found near the 1-to-1 line (**5**). As the new bare ice albedo parameterization allows variations due to atmospheric conditions, as is described in Sect. 2.1.3, some deviations are expected. That is to say,

higher albedos with respect to Rp2 are expected for bare ice, as atmospheric variations usually increase the albedo, e.g., clouds. Additionally, edge errors in Rp2 caused a considerable albedo difference, typically when Rp2 albedo were smaller than 0.35

(**6**). Processes **1** to **6** are all applicable to the region around the K-transect (Fig. 8d), causing a spread for this region that is considerably larger than for the regions in Fig. 8b and Fig. 8c.

The large differences **7** and **8**, that is, all occurrences beyond the dashed lines in Fig. 8a, are not present in the regions considered in Fig. 8b to Fig. 8d. Process **7** occurs almost exclusively next to the ice margin; some in the west, but most in the east and north. In addition to the previously described errors occurring for those points (i.e., mixing with tundra points), snowfall or superimposed ice that is not present in Rp2 can also contribute to the difference. For **8**, most occurrences are associated with area **F** of Fig. 6b and to a lesser extent area **E** and some grid points close to the ice-sheet margin in northern and eastern Greenland. These occurrences represent cases that Rp2 modeled a fresh snow top layer without melt while Rp3 models either bare ice or a melting snow or firn layer.

To conclude, the albedo product of Rp3 is often similar to Rp2, but some distinct differences are still notable (Fig. 6 and Fig. 8). All these differences can be well understood in terms of physical processes.

## 5.2 Clouds

In Rp3, SNOWBAL allows spectral variations in the incoming solar radiation to impact the surface albedo (see Sect. 2.1.2). As clouds absorb mainly in the (infra)red part of the spectrum, a blue shift occurs, which consequently increases the surface albedo (Dang et al., 2015). The cloud dependence of the surface albedo parameterization in Rp2 was limited to the cloud optical thickness, neglecting water vapour while no distinction is made between liquid or ice water clouds. Figure 9 shows for a grid point in south-central Greenland (star in Fig. 6b) the total-sky surface albedo, transmissivity for shortwave radiation in the atmosphere, i.e., the ratio between the top-of-atmosphere (TOA) and surface downwelling radiation, and the fraction of TOA shortwave radiation absorbed in snow as a function of vertically integrated cloud content (VICC), i.e., total liquid water and ice in the atmosphere above a point, for Rp3 and Rp2.

For clear-sky conditions (VICC smaller than 0.05 kg m$^{-2}$), the surface albedo in Rp3 is slightly lower than Rp2 (Fig. 9), which is in agreement with previous results. The surface albedo in Rp3, however, increases more rapidly with VICC than in Rp2, leading to higher albedos for large VICC. Furthermore, the transmissivity decreases slower with VICC than in Rp2 and less radiation is absorbed in snow. For example, for a VICC of 0.3 kg m$^{-2}$, the surface broadband albedo, transmissivity and fraction of absorbed energy changes by 0.02, 0.04 and -0.01, respectively. The latter implies that the net SW absorption decreases by approximately 25%. These differences show that as a cloud thickens and the surface albedo increases in Rp3, more reflected radiation will interact with clouds, eventually raising the transmissivity. Consequently, as these shorter wavelengths now scatter more often and are less likely to be absorbed in the snow, a white-out effect occurs, which is not captured in Rp2.

## 5.3 Albedo seasonality

Seasonal changes of albedo differences in Rp3 with respect to Rp2 are generally small (Fig. 10). During winter (December, January and February (DJF), Fig. 10a), a homogeneous pattern is present with a small negative difference, that is, the albedo of Rp3 is lower than Rp2. This albedo difference can be mostly attributed to spectral albedo effects, and not as much to radiation penetration, as fresh snow layers are thick enough ice-sheet wide for radiation penetration to be negligible. The winter months

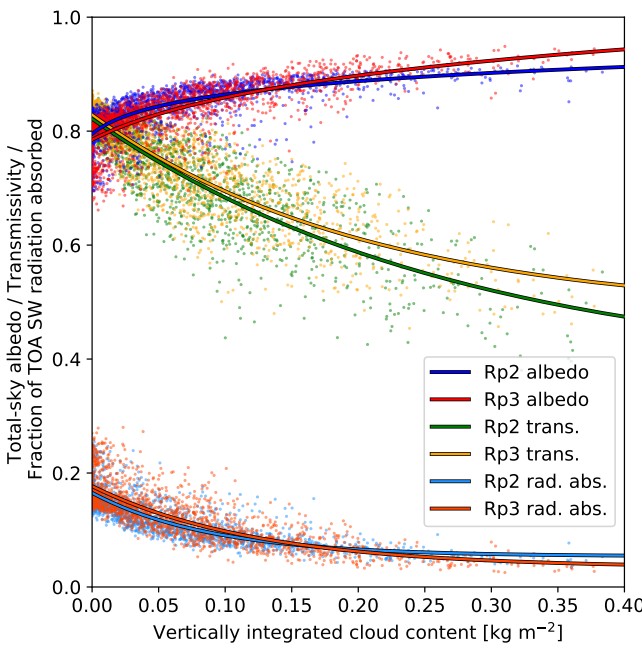

**Figure 9.** Total-sky surface albedo, transmissivity (trans.) and fraction of top-of-atmosphere (TOA) shortwave radiation absorbed in the snowpack (rad. abs.) as a function of the vertically integrated cloud content for a grid point in south-central Greenland, indicated with the star in Fig. 6b, for RACMO2.3p3 (Rp3) and RACMO2.3p2 (Rp2). Every point represents a daily-averaged value between 2006 and 2015 if the mean downward TOA shortwave radiation is larger than 200 W m$^{-2}$. Solid lines show a logarithmic fit for the albedo and an exponential fit for both the transmissivity and the fraction of TOA shortwave radiation absorbed.

are characterized by a large SZA, for which a spectral shift towards longer wavelengths occurs. For a large SZA, the broadband albedo increases for both Rp3 and Rp2. The spectral shift towards longer wavelengths, for which the spectral albedo of IR radiation is low (Fig. 1d), however, is only properly captured in Rp3, mitigating the albedo increase with SZA somewhat (Fig. 11 in Van Dalum et al. (2019)). Consequently, the modeled broadband albedo is lower for Rp3 with respect to Rp2 in the winter months. The red shift in irradiance becomes more dominant towards the northern regions as the SZA increases, explaining the 400   northward albedo difference gradient. Albedo differences diminish as spring progresses (March, April and May (MAM), Fig. 10b) and the sun rises higher in the sky.

    During summer (June, July and August (JJA), Fig. 10c), larger spatial variations are observed, but the albedo differences in the interior remain small. Along the margins, the albedos can be much higher in Rp3 due to tundra contaminated bare ice albedo in Rp2. Around the equilibrium line in the southwest, a line with higher Rp3 albedos is present, which is due to superimposed 405   ice as is described in Sect. 2.1.4. Small positive albedo differences are observed in wet snow regions, e.g., area **D** in Figure 2c, where snow layers with large grains are located close to the surface. Autumn (September, October and November (SON),

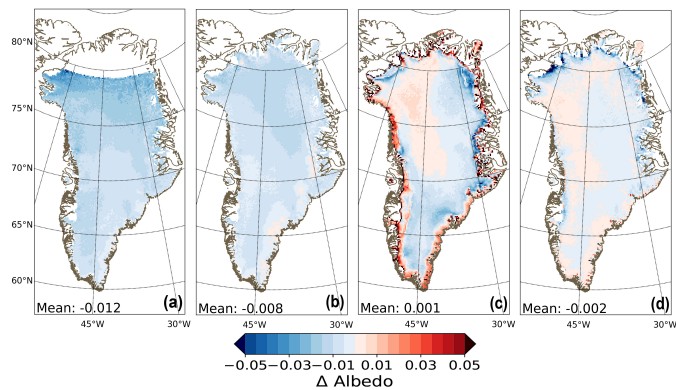

**Figure 10.** Average 16-days running mean total-sky albedo difference between RACMO2.3p3 and RACMO2.3p2 for **(a)** DJF, **(b)** MAM, **(c)** JJA and **(d)** SON, for 15:00 UTC between 2006 and 2015.

Fig. 10d) presents small deviations for most of the ice sheet, as fresh snow layers accumulate and the SZA increases, slowly transitioning to winter conditions and the processes described for DJF become increasingly important.

# 6 Sensitivity experiments: soot concentration

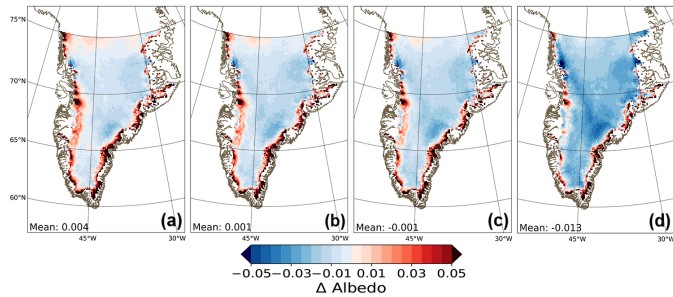

**Figure 11.** Bias-corrected average 16-days running mean albedo difference for 15:00 UTC between RACMO2.3p3 clear-sky albedo and MODIS MCD43A3 CSD albedo product, between 2011 and 2015, for **(a)** an impurity concentration of 0; **(b)** 5 ng g$^{-1}$; **(c)** 10 ng g$^{-1}$ and **(d)** 50 ng g$^{-1}$.

The snow albedo of Rp3 can be tuned with impurity concentration, which in the control run has a fixed value of 5 ng g$^{-1}$. Snow albedo difference for various impurity concentrations with respect to the bias-corrected MODIS CSD albedo is shown in Fig. 11 (2011 - 2015). Excluding all grid points within five grid points of the margin, the mean bias becomes $-0.006$, $-0.009$, $-0.011$ and $-0.022$ for impurity concentrations of 0, 5, 10 and 50 ng g$^{-1}$, respectively. The sensitivity to impurity concentration is generally low except for very high concentrations, e.g., 50 ng g$^{-1}$.

Results for a selection of K-transect and PROMICE observational sites (Fig. 2c) that are sufficiently far away from the ice margin are shown in Fig. 12. Figure 12a is a normalized Taylor diagram (Taylor, 2001) and Fig. 12b shows the bias, misrepresented variability, which is defined as $\sqrt{\mathrm{RMSE}^2 - \mathrm{bias}^2}$, and RMSE, all scaled with the standard deviation in the observations.

In a Taylor diagram, the azimuthal position illustrates the correlation coefficient, the radial distance to the origin the standard deviation and quarter circles with its origin at the 1.0 standard deviation the misrepresented variability. A data set matches the observations perfectly if it is located at the star in Fig. 12a, i.e., with a correlation coefficient and standard deviation of 1 and a misrepresented variability of 0. Data sets located away from the star but on the dashed line have the same variance as the observations, but do not correlate perfectly, leading to a higher misrepresented variability. Data sets close to the origin and close to the y axis, for example, are characterized by an underestimation of the standard deviation and a low correlation coefficient, while data sets beyond the dashed line and close to the x axis overestimate the standard deviation, but have a high correlation coefficient. Similarly, the misrepresented variability is illustrated on the y axis, bias on the x axis and the RMSE on semi circles in Fig. 12b, with data sets performing better close to the origin.

Model performance varies for each observational site, but the sensitivity to small impurity concentrations is generally low. High concentrations of impurities such as 50 ng g$^{-1}$, on the other hand, alter the albedo considerably, reducing the quality of Rp3 for almost all stations. For NUK-U and QAS-U, which are both located in south Greenland and within 50 km of the ice margin, RACMO2 correlates relatively well, but severely underestimates variability and shows a large bias. As the ice margins are characterized by a high soot concentration (Doherty et al., 2010), a higher modeled soot concentrations consequently performs somewhat better for these locations. Snow cover in RACMO2 is also too homogeneous and similar processes that we discussed for area **B** and **D** of Fig. 2c occur. Additionally, RACMO2 is known to overestimate snowfall for QAS-U, inhibiting bare ice to surface (Noël et al., 2018). For S10 and KAN-M, which are most representative for the interior of the ice sheet, the differences between Rp3 with impurity concentration of 5 and 10 ng g$^{-1}$ and to a lesser degree also with 0 ng g$^{-1}$ and Rp2, are small and in good agreement with observations. To summarize, as the sensitivity of Rp3 to small impurity quantities is low and the snow albedo is generally in good agreement with observations, an ice-sheet wide impurity concentration of 5 ng g$^{-1}$ is a safe choice and for the interior is in good agreement with observations (Fig. 11b).

# 7  Summary and conclusions

We evaluated the new spectrally-dependent snow and ice albedo parameterization in RACMO2, based on TARTES and coupled by SNOWBAL, for the Greenland ice sheet. The albedo correlates well with the MODIS MCD43A3 clear-sky diffuse albedo product for both broadband and its seven spectral bands, and performs especially well in the interior. Some discrepancies around the margins are observed, which can be partly ascribed to resolution problems and excessive modeled snowfall, but also to uncertainty in the MODIS product. Around the K-transect, for which many observations are available, the snow and ice albedo in RACMO2 show acceptable small deviations with in-situ and MODIS observations.

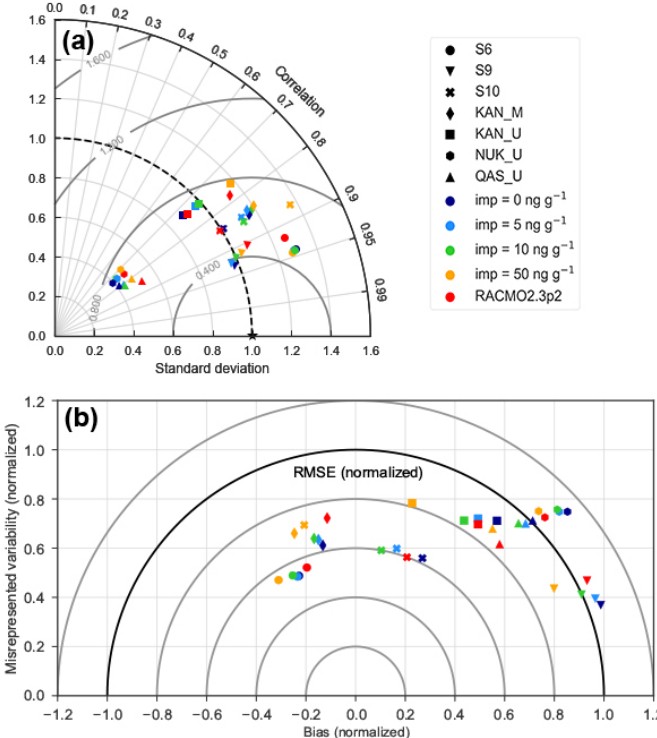

**Figure 12.** (**a**) Normalized Taylor diagram (Taylor, 2001) for 16-days running mean total-sky albedo of RACMO2.3p3 with various impurity concentrations and RACMO2.3p2 with respect to a selection of K-transect and PROMICE in-situ observations for 15:00 UTC between 2011 and 2015. The standard deviation is illustrated by the radial distance to the origin, the correlation coefficient by the azimuthal position and the misrepresented variability, defined as $\sqrt{\mathrm{RMSE}^2 - \mathrm{bias}^2}$, by quarter circles with its origin at the 1.0 standard deviation. (**b**) Extension to the Taylor diagram, which shows the misrepresented variability on the y axis, bias on the x axis and RMSE on semi circles from the origin. All data is normalized with respect to in-situ observations.

With respect to the previous albedo parameterization of RACMO2, slightly lower broadband albedos are modeled. Although large broadband albedo differences at the margins are due to an error in the old version, most changes can be ascribed to improved physics. Radiation penetration, subsurface heating, the inclusion of narrowband albedo and spectral shifts due to solar zenith angle, water vapour and both ice and water clouds are now all incorporated.

There is, however, still room for improvement. The soot concentration for snow is fixed in RAMCO2, while it can change considerably over space and time (Chylek et al., 1992; Doherty et al., 2010; Van Angelen et al., 2012; Dang et al., 2015). Although RACMO2 shows a low sensitivity to small impurity concentrations, a prognostic soot model for snow prescribing a dynamic one-dimensional soot concentration profile is still preferable. Furthermore, no other impurity types are included.

We have also improved the bare ice albedo field by coupling it with TARTES and defining a fictitious grain size, which allows the broadband bare ice albedo to vary with atmospheric conditions. Nevertheless, a proper bare ice albedo module would still be

preferable, as for example, a process like darkening due to biological activity is not incorporated, which may become important for future projections (Tedesco et al., 2016; Tedstone et al., 2019). TARTES is based on simple geometric-optics theory, while an ice albedo module with Mie-scattering theory is required (Gardner and Sharp, 2010).

Evaluation of the narrowband albedo of RACMO2 remains limited, as most of the spectral bands of MODIS are located at the edge of rather large bands of RACMO2 and hence cannot be compared directly. To solve this, we have run TARTES specifically for diffuse radiation for the seven bands of MODIS. For these bands, differences between modeled albedo and observations are low. In the bare ice zone, band 1, 3 and 4, which are within the visible light part of the spectrum, show a larger narrowband albedo bias than the other bands. Larger spatial variations are observed for bands 2 and 5, which are characterized

by a strong sub-band spectral albedo gradient.

       To conclude, the new snow and ice albedo scheme of RACMO2 performs very well compared to remote sensing and in-situ observations for the Greenland ice sheet. Differences with the previous RACMO2 version are generally small, but where differences are observed, the new processes lead to improved broadband albedo estimates. The improvement of the albedo scheme of RACMO2 will enhance its ability to make future climate projections. In a forthcoming publication, we assess the

impact of the new snow and ice albedo scheme on the surface mass balance, surface energy balance and subsurface energy absorption of the Greenland ice sheet.

*Author contributions.*   CTvD, WJvdB and MRvdB started and decided the scope of this study and analyzed the results. CTvD implemented the new snow albedo scheme in the model, performed the simulations and led the writing of the manuscript. SL processed and provided remote sensing data. All authors contributed to discussions on the manuscript.

*Data availability.*   RACMO2.3p3 monthly-averaged data for downward and net shortwave radiation for total-sky and clear-sky conditions at 11 km for Greenland (September 2000-2018) can be found here: https://doi.org/10.5281/zenodo.3763442. More RACMO2.3p3 data are available from the authors.

*Competing interests.*   The authors declare that they have no conflict of interest.

*Acknowledgements.*   We acknowledge financial support from the Netherlands Organization for Scientific Research (NWO) and the ECMWF
for computational time on their supercomputers.

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
