# Peer review of "Evaluation of a new snow albedo scheme for the Greenland ice sheet in the regional climate model RACMO2"

_The Cryosphere, 2020_

## Referee Comment (RC1) · Stephen Warren (Referee) · 26 Jun 2020

Recommendation: Accept after minor revisions.

General comment: This paper documents a revision to the procedure for computing the surface albedo on the Greenland Ice Sheet, for use in a regional climate model. My criticisms are not sufficiently major to require changing the model at this point, but the paper could benefit from better explanations of how solar radiation interacts with snow and ice.

Major comments:

[Figure]

(1) Lines 28-29. "With coarser grains, the likelihood for light to reflect off a grain's surface out of the snowpack reduces, lowering the albedo." This is not true; reflection off a grain's surface is independent of grain size and is anyway a minor contributor to the albedo; successive refraction instead dominates (Bohren and Barkstrom, 1974). The correct explanation for dependence of albedo on grain size is given by Warren (2019, cited in the paper): "In coarse-grained snow, a photon travels a longer distance through ice between opportunities for scattering than in fine-grained snow, so it is more likely to be absorbed, and therefore a snowpack of coarse grains has lower albedo."

(2) Lines 140-142. "We assume that clean bubble-free ice has an albedo of approximately 0.6 (Reijmer et al., 2001) and that the bare ice albedo is subsequently lowered by . . . bubbles . . . ." This statement is wrong in three ways. First, Reijmer et al. did not say that bubble-free ice has albedo 0.6. Second, a thick layer of ice, if it is really bubble-free, will have very low albedo, about 0.07. Bubble-free ice does sometimes occur on frozen lakes, where it is appropriately called "black ice". Third, the albedo of ice is raised by bubbles, not lowered. The bubble surfaces are what are responsible for the model's assumed specific surface area of 0.788 m2/kg. Without bubbles (or cracks), the SSA of thick ice would be zero, and the albedo would be only the Fresnel reflection at the top surface (which is 0.07 for diffuse incidence). Figure 17b of Dadic et al. (2013, cited in the paper) shows the broadband albedo versus SSA for ice, firn, and snow, all on the same plot.

(3) Superimposed ice is mentioned in several places (e.g. lines 285-286), but is not discussed adequately. The reader wants to know the definition of superimposed ice, and why its albedo is higher than that of bare glacier ice. Glacier ice is formed by compression of snow under pressure, whereas superimposed ice is formed by refreezing of meltwater. The paper describes superimposed ice as having a granular structure with grain radius 0.7-1.0 mm, which could result in a higher SSA than in glacier ice. But there are reports of such a surface granular layer developing also in glacier ice exposed to sunlight, as in Figure 1 of Mueller and Keeler (1969).

(4) The effect of a thin snow layer on top of ice is belittled in the Rp3 model. On Line 329, Location 8, with an Rp3 albedo of only 0.45, is said to be thin snow on bare ice. But in a comparable situation, Figure 2 of Brandt et al. (2005) showed that when thick bare sea ice of albedo 0.49 was covered with 5-10 mm of snow, its albedo rose to 0.81. So why was the Rp3 albedo so low? You might check to see whether the snow was melting.

Minor comments:

Line 32. In spite of the importance of snow grain shape for albedo, a model of spherical grains still accurately reproduces the measured spectral albedo, by adjusting the grain radius. The reason this procedure works was explained by Dang et al. (2016).

Line 38. "Rayleigh scattering [by the atmosphere] is more effective for larger wavelengths . . . ". Change "larger" to "shorter". Rayleigh scattering is inversely proportional to the fourth power of the wavelength, so blue is scattered more than red.

Line 42. Change "incoming radiation" to "incoming direct-beam radiation".

Line 54. "version 2.3p3". I am guessing that the "p" means "polar". Is that correct?

Line 64. "the new snow albedo parameterization is used to update the ice albedo scheme". Does this mean that snow and ice both use the same parameterization, just with different coefficients?

Lines 77-78. "The polar version of RACMO2 . . . is adapted for glaciated regions by using a multilayer snowpack . . ." Is this scheme used also for seasonal snow in midlatitudes, or only for polar latitudes? If the latter, what is the latitude domain to which the model is applied?

Lines 97-98. "For ice layers . . . only a density reduction takes place and no thinning occurs." This is backwards. When ice melts, it thins, but the density of the remaining ice is unchanged.

Line 104 (and elsewhere). "Rp2 uses . . ." To distinguish Rp2 and Rp3, I suggest using past-tense for Rp2 and present-tense for Rp3.

Line 114. "grain shape . . . are all explicitly resolved". How is grain shape described in the model?

Lines 145-149. Bohren (1983) was the first one to show that bubbly ice could be modeled as very-coarse-grained snow, as done here.

Line 161. "unrealistically high albedos of 0.7". A reference is needed here for measured albedos of superimposed ice, to show that an albedo of 0.7 is too high.

Line 170, reference to Chylek et al. A better reference is McConnell et al. (2007).

Line 171. It is true that Doherty et al. (2010) found an average of only 3 ng/g soot, but as the snow melts the soot accumulates at the surface, attaining concentrations in the top centimeter of 20-100 ng/g at Dye-2 (Figure 8a of Doherty et al. 2013).

Lines 178-179. "the bare ice albedo value is replaced with the bare ice soot concentration . . ." Rewrite this sentence. It does not make sense to replace an albedo value with a soot concentration; they have different units.

Lines 233-234. "The region indicated by C . . . albedo difference during winter . . ." Region C is at latitude 72.5 degrees, so on the March equinox at noon the SZA is 72.5 degrees, and throughout the winter the SZA is greater than this. Since you are limiting your comparisons to SZA<55 degrees, in winter the Sun is too low for any comparisons.

Lines 252-253. "metamorphosis . . . metamorphism". Use consistent terminology. LaChapelle favored "metamorphism".

Lines 269-270. "large quantities of soot . . . not enough to lower the albedo for bare ice to MODIS values". Bare ice on Greenland contains not just soot but also dust, cryoconite, and algae, all of which reduce the albedo. Bøggild et al. (2010, Table 1)

reported ice albedos as low as 0.2.

Line 335 Section 5.2. In this section, clarify that by "albedo" you mean surface albedo, not TOA albedo.

Line 356. Change "larger" to "longer".

Line 366. "[In autumn] the SZA is too small to make a significant difference". The Sun is nearly as low in SON as in DJF; they are the two "low-sun" seasons. So what you said about the effect of SZA in DJF should also apply to SON.

Line 417 states that the TARTES snow model is based on Rayleigh scattering. Rayleigh scattering applies to gases, and to particles much smaller than the wavelength. Maybe TARTES uses Rayleigh theory for submicron soot particles, but surely not for snow grains.

Figure 1. Much of the coastal region, but not all, is blank. What are the blank areas; are these ice-free and snow-free?

Figure 1 caption. Line 1. "16-day". What months are included? Lines 1-2. "The albedo is limited between 0.30 and 0.55". I don't understand this; you know that the accumulation area has albedo >0.55. Line 3. "impurities". What kind of impurity is assumed, and what is its mass-absorption cross-section?

Figure 2. Put labels on the latitude & longitude lines (also on Figure 1).

Figure 3 caption, line 4. "bin size is 0.01". What units?

Figure 9 caption, line 1. "Total-sky albedo". Clarify that you mean surface albedo, not top-of-atmosphere albedo.

References:

Bøggild, C.E., R.E. Brandt, K.J. Brown, and S.G. Warren, 2010: The ablation zone in northeast Greenland: Ice types, albedos, and impurities. J. Glaciol., 56, 101-113.

[Figure]

Bohren, C.F., 1983: Colors of snow, frozen waterfalls, and icebergs. J. Opt. Soc. Am., 73, 1646-1652.

Bohren, C.F., and B.R. Barkstrom, 1974: Theory of the optical properties of snow. J. Geophys. Res., 79, 4527-4535.

Brandt, R.E., S.G. Warren, A.P. Worby, and T.C. Grenfell, 2005: Surface albedo of the Antarctic sea-ice zone. J. Climate, 18, 3606-3622. Dang, C., Q. Fu, and S.G. Warren, 2016: Effect of snow grain shape on snow albedo. J. Atmos. Sci., 73, 3573-3583. doi:10.1175/JAS-D-15-0276.1

Doherty, S.J., T.C. Grenfell, S. Forsström, D.L. Hegg, R.E. Brandt, and S.G. Warren, 2013: Observed vertical redistribution of black carbon and other insoluble light-absorbing particles in melting snow. J. Geophys. Res., 118, doi:10.1002/jgrd.50235.

McConnell, J.R., R. Edwards, G.L. Kok, M.G. Flanner, C.S. Zender, E.S. Saltzman, J.R. Banta, D.R. Pasteris, M.M. Carter, and J.D.W. Kahl, 2007: 20th century industrial black carbon emissions altered Arctic climate forcing. Science, 317, 1381-1384, doi:10.1126/science.1144856.

Mueller, F., and C.M. Keeler, 1969: Errors in short-term ablation measurements on melting ice surfaces. J. Glaciol., 8, 91-105.

---

## Referee Comment (RC2) · Mark Flanner (Referee) · 29 Jun 2020

This study incorporates the TARTES and SNOWBAL radiative transfer schemes into RACMO2, and evaluates the performance of the new model against remote sensing and in-situ measurements, and also against the old RACMO2 scheme. The evaluation shows general improvement in comparison with the original scheme, and is also more justifiable on physical grounds, including more realistic treatment of spectral characteristics and vertical penetration of radiation. The use of soot as a tuning parameter to achieve targeted albedo is not a desirable long-term solution, but okay in the context presented. Overall, the paper is well-written, well-presented, and scientifically sound.

[Figure]

Several minor issues outlined below should be clarified, however, before publication in TC.

General issues:

The MODIS albedo product used for model evaluation is Version 6 of MCD43A3. The version evaluated by Stroeve et al (2013), however, was version 5. Hence the RMSE and biases reported for MCD43A3, and used to tune the model albedo, may not be applicable. I am unsure of changes in the retrieval algorithm between versions 5 and 6, but they may be non-negligible (see, e.g., Polashenski et al., 2015, doi:10.1002/2015GL065912). Some exploration and assessment of this issue should be included.

Section 2.1.1 - Are the multilayer firn updates new features that need to be introduced here, or are/can they be described in another study? I ask because this sub-section seems somewhat tangential to the study, which otherwise focuses on snow albedo.

Minor issues:

Lines 47-52: Some models do conduct coarsely-resolved spectral calculations. For example, The CESM and E3SM models include SNICAR, which currently calculates snow albedo in 5 spectral bands when embedded in these GCMs. Insolation from the atmosphere is partitioned into only 2 bands (visible and near-IR), however. SNICAR also represents sub-surface absorption of solar energy. Details can be found in the CLM technical note: http://www.cesm.ucar.edu/models/cesm2/land/CLM50_Tech_Note.pdf

line 101-102: Why was the initialized ice density changed from 910 to 917 kg/m3, given the next sentence which states that bare ice density is usually lower than 917 kg/m3? Also, does the ice density change with time in the model? Finally, "mimicking" might be better replaced with "indicating", in this context.

line 118-129: Description of SNOWBAL: It would be helpful to mention or briefly describe how much of an impact on broadband albedo/absorption this clever selection of

sub-band wavelength causes relative to use of the sub-band center wavelength, which is the technique likely employed by most others.

line 125-129, and Figure 1: Is the MCD43 "clear-sky diffuse" albedo field equivalent to their "white-sky" albedo? If so, I suggest applying consistent terminology throughout the paper. Also, is the only difference between your clear-sky and cloudy diffuse albedo fields associated with cloud-induced spectral shifts? I assume the clear-sky diffuse albedo only minimally impacts the clear-sky albedo, except at very short wavelengths where Rayleigh scattering is appreciable.

line 140: "Firstly, we assume that clean bubble-free ice has an albedo of approximately 0.6" - Pure, bubble-free ice technically has a much lower albedo than this, as described by the Fresnel equations. I assume this higher (measured?) albedo is caused by surface scattering and roughness. If so, this should be mentioned.

line 154: What type of impurity is indicated with these concentrations? (Presumably soot).

line 169: "RACMO2 only allows for a fixed soot concentration." - But to be clear, the prescribed soot concentration varies over bare ice, and is only fixed over snow, correct? This distinction could use some clarification.

line 184-185: "But note that the MCD43A3 WSA product remains a slightly different albedo product than the clear-sky RACMO2 albedo it evaluates, which includes both direct and diffuse radiation." - As described earlier, however, both direct and diffuse clear-sky albedos are calculated by the model. Why not use the diffuse clear-sky albedo for comparison with MCD43 white-sky albedo. Wouldn't this be an apples-for-apples comparison?

line 199: Again, I believe Stroeve et al (2013) use version of 5 of this product.

line 357: "... high SZA... The spectral albedo of IR radiation is low, hence the broad-band albedo drops" - But the SZA grazing effect outweighs the spectral shift, producing

*higher* albedo at higher SZA, doesn't it? Your language on p.2 suggests so: "... this increase of spectral albedo at large SZA is largely mitigated by the red shift..." (i.e., largely, but not entirely, mitigated).

---

## Author Comment (AC1) · 16 Jul 2020

Referee comment response on the manuscript: Evaluation of a new snow albedo scheme for the Greenland ice sheet in the regional climate model RACMO2 by C.T. van Dalum et al.

We would like to thank the reviewers for their constructive comments that have improved the accuracy of the evaluation and the clarity of the paper. For a better version of this comment response using colors for the original comment, our response and the changes that we would implement in the manuscript, see the supplement.

Review #1 by Stephen Warren (1) Lines 28-29. "With coarser grains, the likelihood for light to reflect off a grain's surface out of the snowpack reduces, lowering the albedo." This is not true; reflection off a grain's surface is independent of grain size and is anyway a minor contributor to the albedo; successive refraction instead dominates (Bohren and Barkstrom, 1974). The correct explanation for dependence of albedo on grain size is given by Warren (2019, cited in the paper): "In coarse-grained snow, a photon travels a longer distance through ice between opportunities for scattering than in fine-grained snow, so it is more likely to be absorbed, and therefore a snowpack of coarse grains has lower albedo." As our description was indeed inaccurate, we changed the following: Page 2: With coarser grains, light has to travel longer through ice before it has the opportunity to reflect off a grain's surface out of the snowpack than for fine-grained snow, hence lowering the albedo (Wiscombe and Warren, 1980; Gardner and Sharp, 2010; Picard et al., 2012, Warren, 2019). Fresh snow with a small grain radius...

(2) Lines 140-142. "We assume that clean bubble-free ice has an albedo of approximately 0.6 (Reijmer et al., 2001) and that the bare ice albedo is subsequently lowered by . . . bubbles . . . ." This statement is wrong in three ways. First, Reijmer et al. did not say that bubble-free ice has albedo 0.6. Second, a thick layer of ice, if it is really bubble-free, will have very low albedo, about 0.07. Bubble-free ice does sometimes occur on frozen lakes, where it is appropriately called "black ice". Third, the albedo of ice is raised by bubbles, not lowered. The bubble surfaces are what are responsible for the model's assumed specific surface area of 0.788 m2/kg. Without bubbles (or cracks), the SSA of thick ice would be zero, and the albedo would be only the Fresnel reflection at the top surface (which is 0.07 for diffuse incidence). Figure 17b of Dadic et al. (2013, cited in the paper) shows the broadband albedo versus SSA for ice, firn, and snow, all on the same plot. You are right. Therefore we changed the following: Page 5: Firstly, we assume that clean blue ice has an albedo of approximately 0.6 (Reijmer et al, 2001; Dadic et al, 2013). Blue ice is typically found in areas with a very smooth surface and high sublimation rates, but no melt, and has a high bubble content, leading

**TCD**
to a relatively high albedo. The bare ice albedo is subsequently lowered by standing water, surface roughness and impurities. Furthermore, we assume that MODIS...

(3) Superimposed ice is mentioned in several places (e.g. lines 285-286), but is not discussed adequately. The reader wants to know the definition of superimposed ice, and why its albedo is higher than that of bare glacier ice. Glacier ice is formed by compression of snow under pressure, whereas superimposed ice is formed by refreezing of meltwater. The paper describes superimposed ice as having a granular structure with grain radius 0.7-1.0 mm, which could result in a higher SSA than in glacier ice. But there are reports of such a surface granular layer developing also in glacier ice exposed to sunlight, as in Figure 1 of Mueller and Keeler (1969). We agree that we have not clarified the definition of superimposed ice clearly. Exactly as you have stated, we mean with superimposed ice layers that have refrozen meltwater in it, which is different from bare ice that is formed by compaction. More importantly for us, however, is the different way that superimposed ice is treated compared to bare ice in the model. Superimposed ice does not use the bare ice albedo scheme, as is described in Sect. 2.1.3, but uses the snow albedo scheme and the changes described in Sect. 2.1.4. We think that changing the following in Sect. 2.1.4 is sufficient for the reader to know what we mean with superimposed ice for the rest of the paper. Page 6: In Rp3, superimposed ice is treated differently than glacial ice. Superimposed ice forms in snow layers by refreezing of melt water, while glacial ice forms by compaction of snow. As superimposed ice has a granular structure (Granskog et al., 2006), it has to be treated differently than bare ice. Due to the granular structure, it is desirable to use the snow albedo scheme of Sect. 2.1.2 over the bare ice albedo scheme of Sect. 2.1.3, as a fixed rather large grain radius is used for bare ice. However, without additional corrections, the typical grain size of model layers with superimposed ice in Rp3 is 0.7 to 1.0 mm, leading to unrealistically high albedos of approximately 0.7. The typical albedo of superimposed ice is 0.65, as is measured by Knap and Oerlemans (1996) at S9 of the K-transect. In order to improve this, a minimum grain radius for superimposed ice is imposed, which increases linearly from 0.720 mm for a density of 750 kg m3, to the

**TCD**
bare ice value of 4.152 mm for a density of 917 kg m3. This superimposed ice layer uses the same impurity concentration as a snow layer. This correction leads to realistic albedos for superimposed ice and exposed ice lenses.

(4) The effect of a thin snow layer on top of ice is belittled in the Rp3 model. On Line 329, Location 8, with an Rp3 albedo of only 0.45, is said to be thin snow on bare ice. But in a comparable situation, Figure 2 of Brandt et al. (2005) showed that when thick bare sea ice of albedo 0.49 was covered with 5-10 mm of snow, its albedo rose to 0.81. So why was the Rp3 albedo so low? You might check to see whether the snow was melting. After analysis of these cases, we agree that our conclusion was not correct. Location 8 represents grid points where Rp3 models either bare ice or melting snow or firn, while Rp2 models a fresh snow top layer. So it does not show a strong direct radiation penetration effect on albedo. We have changed the following: Page 13: At the end of the ablation season when new snow layers start to form, melt can still occur, forming melt water within those layers. This is more common in Rp3, as internal heating increases the subsurface temperature and therefore more melt occurs, changing the snow structure and lowering the albedo in Rp3 with respect to Rp2. Furthermore, if melt is strong enough, it can remove or thin the fresh snow top layer, exposing bare ice underneath. As the accumulation season... Page 15: ... and to a lesser extent area E and some grid points close to the ice-sheet margin in northern and eastern Greenland. These occurrences represent cases that Rp2 models a fresh snow top layer without melt while Rp3 models either bare ice or a melting snow or firn layer.

Minor comments Line 32. In spite of the importance of snow grain shape for albedo, a model of spherical grains still accurately reproduces the measured spectral albedo, by adjusting the grain radius. The reason this procedure works was explained by Dang et al. (2016). The reviewer is right and we have added the following in the manuscript. Page 2: ...out of the snowpack (Libois et al., 2013; He et al., 2018a), but Dang et al. 2016 show that a model with spherical grains can still accurately reproduce the measured spectral albedo by adjusting the grain radius. To summarize, it is thus essential...
Line 38. "Rayleigh scattering [by the atmosphere] is more effective for larger wavelengths...". Change "larger" to "shorter". Rayleigh scattering is inversely proportional to the fourth power of the wavelength, so blue is scattered more than red. Changed as requested Page 2: ...as Rayleigh scattering by the atmosphere is more effective for shorter wavelengths...

Line 42. Change "incoming radiation" to "incoming direct-beam radiation". Done Page 2: ...of the incoming direct-beam radiation...

Line 54. "version 2.3p3". I am guessing that the "p" means "polar". Is that correct? Yes, "p" means polar. To clarify this, we have added the following: Page 2: The polar (p) version of RACMO2 is a model developed..

Line 64. "the new snow albedo parameterization is used to update the ice albedo scheme". Does this mean that snow and ice both use the same parameterization, just with different coefficients? Snow and ice now both use the TARTES/SNOWBAL albedo parameterization, but the grain radius/SSA and impurity content for bare ice is different. Page 3: Additionally, the new snow albedo parameterization is used to develop a new ice albedo scheme.

Lines 77-78. "The polar version of RACMO2 . . . is adapted for glaciated regions by using a multilayer snowpack . . ." Is this scheme used also for seasonal snow in midlatitudes, or only for polar latitudes? If the latter, what is the latitude domain to which the model is applied? Multilayer snowpack is not used for seasonal snow in midlatitudes, it is only used for glaciated tiles, which is for our domain the Greenland ice sheet and its surrounding glaciers, if those glaciers are large enough to fill up a grid point. Page 3: ... adapted for glaciated tiles by using ...

Lines 97-98. "For ice layers . . . only a density reduction takes place and no thinning occurs." This is backwards. When ice melts, it thins, but the density of the remaining ice is unchanged. Paper Although you are right that the density of the remaining ice will remain, the density of the model layer will go down as pore space is created. Moreover,
if melting ice reaches the surface, it will thin the layer. To clarify this, we changed the following: Page 4: Thirdly, internal energy absorption heats subsurface snow layers and can induce melt. In Rp3, melt will only thin a subsurface snow layer, i.e., a layer with a density below 700 kg m-3 and not change its density. For ice layers, i.e., with a layer density larger than 830 kg m-3, melt creates pore space, reducing the layer density and no thinning occurs.

Line 104 (and elsewhere). "Rp2 uses . . ." To distinguish Rp2 and Rp3, I suggest using past-tense for Rp2 and present-tense for Rp3. As requested, we changed the tense to the past on relevant places when Rp2 is used.

Line 114. "grain shape . . . are all explicitly resolved". How is grain shape described in the model? In the model, all grains are assumed to be spherical. Page 4: . . . are all explicitly resolved. In this study, all grains are spherically shaped. TARTES is able to calculate. . .

Lines 145-149. Bohren (1983) was the first one to show that bubbly ice could be modeled as very-coarse-grained snow, as done here. Thank you for this suggestion. We have added a reference to the work of Bohren (1983) in the manuscript. Page 5: ...and thus can be used to indicate clean, bare ice, which is similar to the findings of Bohren (1983).

Line 161. "unrealistically high albedos of 0.7". A reference is needed here for measured albedos of superimposed ice, to show that an albedo of 0.7 is too high. As is described in comment (3) of this review, we have added the following: Page 6: The typical albedo of superimposed ice is 0.65, as is measured by Knap and Oerlemans (1996) at S9 of the K-transect. In order to improve this...

Line 170, reference to Chylek et al. A better reference is McConnell et al. (2007). Changed as requested

Line 171. It is true that Doherty et al. (2010) found an average of only 3 ng/g soot, but
as the snow melts the soot accumulates at the surface, attaining concentrations in the top centimeter of 20-100 ng/g at Dye-2 (Figure 8a of Doherty et al. 2013). We have specified where this 3 ng/g is a representative number. We chose to have the impurity content right for non-melting conditions, in order to get the melt-onset timing right. Page 7: In the interior and when no melt occurs, the soot concentration is approximately 3 ng g-1...

Lines 178-179. "the bare ice albedo value is replaced with the bare ice soot concentration . . ." Rewrite this sentence. It does not make sense to replace an albedo value with a soot concentration; they have different units. We have adjusted the formulation as the initial formulation resembled the numerical implementation of boundary files, while for the paper this is indeed only confusing. Page 7: . . . initialization day, i.e., 1 September 2000, but the fresh snow sub-layer data is omitted. Page 7: Furthermore, Rp3 is initialized with a soot concentration that can be used to calculate the bare ice albedo (Sect. 2.1.3).

Lines 233-234. "The region indicated by C . . . albedo difference during winter . .." Region C is at latitude 72.5 degrees, so on the March equinox at noon the SZA is 72.5 degrees, and throughout the winter the SZA is greater than this. Since you are limiting your comparisons to SZA

Lines 252-253. "metamorphosis . . . metamorphism". Use consistent terminology. LaChapelle favored "metamorphism". Changed as requested

Lines 269-270. "large quantities of soot . . . not enough to lower the albedo for bare ice to MODIS values". Bare ice on Greenland contains not just soot but also dust, cryoconite, and algae, all of which reduce the albedo. Bøggild et al. (2010, Table 1) reported ice albedos as low as 0.2. We have adjusted the text to include this notion. Page 12: It is, however, expected that Rp3 needs large amounts of soot to lower the bare ice albedo, as no dust, cryoconite and algae are modeled, all of which lower the albedo (Bøggild et al., 2010).

Line 335 Section 5.2. In this section, clarify that by "albedo" you mean surface albedo, not TOA albedo. As requested, we changed the term 'albedo' to 'surface albedo' in this section.

Line 356. Change "larger" to "longer". Changed as requested.

Line 366. "[In autumn] the SZA is too small to make a significant difference". The Sun is nearly as low in SON as in DJF; they are the two "low-sun" seasons. So what you said about the effect of SZA in DJF should also apply to SON. You are right, and we change it accordingly. Page 18: Autumn (September, October and November (SON), Fig 10d) presents small deviations for most of the ice sheet, as fresh snow layers accumulate and the SZA increases, slowly transitioning to winter conditions and the processes described for DJF become increasingly important.

Line 417 states that the TARTES snow model is based on Rayleigh scattering. Rayleigh scattering applies to gases, and to particles much smaller than the wavelength. Maybe TARTES uses Rayleigh theory for submicron soot particles, but surely not for snow grains. TARTES uses a geometric-optics method, so not Rayleigh scattering, so you are right. We have changed the following: Page 4: ...asymptotic analytical radiative transfer theory (Kokhanovsky, 2004; He and Flanner, 2020) using the geometric-optics method... Page 5: Despite not using Mie-scattering theory, the spectral curve

TCD
in TARTES for this SSA value resembles the expected curve for bare ice... Page 19: TARTES is based on simple geometric-optics theory, while an ice albedo module with Mie-scattering theory is...

Figure 1. Much of the coastal region, but not all, is blank. What are the blank areas; are these ice-free and snow-free? That is correct, the blank areas are not glaciated. For the changes made in the text: see next comment

Figure 1 caption. Line 1. "16-day". What months are included? Lines 1-2. "The albedo is limited between 0.30 and 0.55". I don't understand this; you know that the accumulation area has albedo >0.55. Line 3. "impurities". What kind of impurity is assumed, and what is its mass-absorption cross-section? As requested, we added some missing information and changed the caption of Figure 1 to: Page 6: Figure 1 (a) Lowest five percent of the MODIS MCD43A3v5 1 km 16-day clear-sky diffuse albedo field for glaciated areas for the period 2000-2015. As this albedo field is used to determine a bare ice albedo field, it is limited between 0.30 for dark ice in the ablation zone, and 0.55 in the accumulation zone under perennial snow for consistency with RACMO2 (Noël et al., 2018). (b) Bare ice impurity field that is implemented in RACMO2.3p3 for glaciated grid points. Here, all impurities are soot. (c) In blue, fitting the specific surface area (SSA) to clean blue ice albedo, which is assumed to be 0.6. The fitted SSA equals 0.788 m2 kg-1. In red, the soot concentration as a function of albedo required to successfully convert the MODIS albedo field into an impurity field. For both lines, clear-sky conditions are assumed for a SZA of 60o and RACMO2 irradiance profiles are used to convert narrowband to broadband albedo. (d) Spectral albedo for clean fresh snow (in black), and for an ice profile with the fitted SSA of 0.788 m2 kg-1 for various impurity concentrations. The first twelve spectral bands of RACMO2 are indicated by vertical dotted lines and black numbers. Red bars and numbers indicate the seven MODIS spectral bands. The albedo for the cases with soot concentrations of 0.2 and 1.5 mu g g-1 are indicated with corresponding colored dots in (c). Page 5: Hence, albedos ranging from 0.30 to 0.55 observed for the GrIS are obtained by increasing the soot TCD
content, with the absorption cross section for soot that is determined by Kokhanovsky (2004)

Figure 2. Put labels on the latitude & longitude lines (also on Figure 1). As requested, we added latitude and longitude labels on all Figures with maps.

Figure 3 caption, line 4. "bin size is 0.01". What units? The bin size has no unit, as the figure shows the RACMO2.3p3 clear-sky albedo compared to MODIS clear-sky diffuse albedo.

Figure 9 caption, line 1. "Total-sky albedo". Clarify that you mean surface albedo, not top-of-atmosphere albedo. Changed accordingly. Č Review #2 by Mark Flanner (1) The MODIS albedo product used for model evaluation is Version 6 of MCD43A3. The version evaluated by Stroeve et al (2013), however, was version 5. Hence the RMSE and biases reported for MCD43A3, and used to tune the model albedo, may not be applicable. I am unsure of changes in the retrieval algorithm between versions 5 and 6, but they may be non-negligible (see, e.g., Polashenski et al., 2015, doi:10.1002/2015GL065912). Some exploration and assessment of this issue should be included. You are right, some of the references regarding MODIS evaluation used version 5 instead of version 6. We investigated the topic a bit more, and various studies are performed evaluating version 6 for the Greenland ice sheet, some even comparing it to version 5 (e.g., Wright et al., 2014; Burkhart et al., 2107; Moustafa et al., 2017; Wang et al., 2018, which are all included now in the manuscript). Wright et al. (2014) state that for Summit, the mean albedo difference of MODIS Version 6 with in-situ observations is 0.015 and the RMSE is 0.026, that is, on average a slightly overestimation of the MODIS broadband albedo with respect to observations. We have changed Figure 2d and Figure 11 accordingly, and changed the following: Page 1: ... leading to a negligible domain-averaged broadband albedo bias for the interior. Page 7: ...using the MCD43A3 Version 006 Albedo Model daily dataset using 16-day Terra and Agua MODIS data for white-sky, i.e., clear-sky diffuse (CSD), and black-sky, i.e., clear-sky direct (CSDir), conditions (Schaaf and Wang, 2015). Page 8: ... but now after applying
a uniform -0.015 bias correction to the MODIS MCD43A3 data. Page 8: ...some regions have limited coverage. Extensive evaluation of the MCD43A3 Version 006 Albedo product shows that it compares well with observations (Wright et al., 2014; Burkhart et al., 2107; Moustafa et al., 2017; Wang et al., 2018). For Summit located in central Greenland, Wright et al. (2014) report a RMSE and mean albedo difference with respect to in-situ observations of 0.026 and 0.015, respectively, indicating that MCD43 slightly overestimates the albedo. Page 9: ...we observe an average bias of -0.022, which is close to the mean difference of -0.015 for Summit reported by Wright et al. (2014). Correcting for this MCD43 mean albedo difference (Fig. 2d), the bias for area A reduces to -0.007, supporting excellent... Page 10: ...of the MODIS CSD albedo product (Wright et al., 2014). Furthermore, ... Page 18: Snow albedo difference for various impurity concentrations with respect to the bias-corrected MODIS CSD albedo is shown in Fig. 11 (2011 - 2015). Excluding all grid points within five grid points of the margin, the mean bias becomes -0.006, -0.009, -0.011 and -0.022 for impurity concentrations...

(2) Section 2.1.1 - Are the multilayer firn updates new features that need to be introduced here, or are/can they be described in another study? I ask because this subsection seems somewhat tangential to the study, which otherwise focuses on snow albedo. Although I can understand that it feels out of place here, the updated RACMO version includes these changes, so they have to be mentioned. Moreover, as running RACMO is computationally quite expensive, we decided that we run RACMO with the final version that includes these changes. As these changes impact the structure of the snow pack, especially in the vertical resolution, they therefore also impact the albedo and have to be mentioned. Unfortunately, these are on itself not enough to publish about. To clarify this a bit more, we added the following: Page 3: In Rp3, the multilayer firn module has been rewritten to improve code efficiency and reduce numerical diffusion. As the surface albedo depends on the structure of the snowpack, any changes made to the multilayer firn module are therefore also important to discuss. The update of this module consists of four modifications.

**TCD**
Minor comments Lines 47-52: Some models do conduct coarsely-resolved spectral calculations. For example, The CESM and E3SM models include SNICAR, which currently calculates snow albedo in 5 spectral bands when embedded in these GCMs. Insolation from the atmosphere is partitioned into only 2 bands (visible and near-IR), however. SNICAR also represents sub-surface absorption of solar energy. Details can be found in the CLM technical note: http://www.cesm.ucar.edu/models/cesm2/land/CLM50\_Tech\_Note.pdf The reviewer is right and we added the following to clarify this: Page 2: RCMs and GCMs commonly perform their radiative calculations for the atmosphere on a limited number of spectral bands. The albedo of such a spectral band is defined as the narrowband albedo. Some of these models do conduct coarsely-resolved spectral calculations on a few bands, like E3SM and CESM (Caldwell et al., 2019; Danabasoglu et al., 2020), but more often they do not use narrowband albedos and determine a broadband albedo instead, bypassing its spectral bands and neglecting any spectral albedo variations.

line 101-102: Why was the initialized ice density changed from 910 to 917 kg/m3, given the next sentence which states that bare ice density is usually lower than 917 kg/m3? Also, does the ice density change with time in the model? Finally, "mimicking" might be better replaced with "indicating", in this context. The ice density is initialized with 917 kg/m3 to better match pure ice density, which is better to convert the effective grain radius into a SSA. A layer that consists of ice usually has a considerably lower density, however, as pore space and air bubbles are created over time. So even though the ice itself has a density of 917 kg/m3, the layer where it is located usually does not. We change the following to clarify this. We also changed the 'mimic' term on various places. Page 4: Finally, the initialized ice density is increased from 910 kg m-3 to 917 kg m-3, which is more in agreement with observations (Bader, 1964), and is used to convert the effective grain radius into a SSA. Furthermore, as ice layers melt, pore space is created, which lowers the layer density. The lower density for bare ice layers then indicates that air bubbles are present within the ice. Page 5: ...impurity concentration field to be used for bare ice albedo calculations to resemble
the broadband MODIS albedo.

line 118-129: Description of SNOWBAL: It would be helpful to mention or briefly describe how much of an impact on broadband albedo/absorption this clever selection of sub-band wavelength causes relative to use of the sub-band center wavelength, which is the technique likely employed by most others. As requested, we added the following to illustrate the impact that the use of 'representative wavelengths' has on the albedo product. Page 4: ...narrowband albedos by TARTES. Using simply the wavelength of the center of the spectral bands increases the root-mean-square error (RMSE) of the broadband albedo by approximately 0.05 and 0.04 for clear-sky direct and clearsky diffuse radiation, respectively, and increases even more for cloudy conditions (Van Dalum et al., 2019). The representative wavelength...

line 125-129, and Figure 1: Is the MCD43 "clear-sky diffuse" albedo field equivalent to their "white-sky" albedo? If so, I suggest applying consistent terminology throughout the paper. Also, is the only difference between your clear-sky and cloudy diffuse albedo fields associated with cloud-induced spectral shifts? I assume the clear-sky diffuse albedo only minimally impacts the clear-sky albedo, except at very short wavelengths where Rayleigh scattering is appreciable. From what I understand, the clear-sky diffuse albedo of MCD43 is equivalent to their 'white-sky' albedo. The reason we chose to stick with the term 'white-sky' albedo when speaking about MODIS is albedo is to be consistent with their terminology. But as requested, we changed the terminology of 'white-sky' and 'black-sky' albedo to clear-sky diffuse (CSD) and clear-sky direct (CSDir) albedo on various places in the manuscript. The most important changes are highlighted here, and Fig 3 and 5 have been changed accordingly. Page 5: In this manuscript, 'albedo' without further specification refers to the broadband albedo. Clear-sky direct (CSdir) and clear-sky diffuse (CSD) albedo refers to surfaces illuminated only by direct radiation or diffuse radiation, respectively. Combined, they are referred to clear-sky albedo. The clear-sky and cloudy-sky albedo can in turn be combined to a total-sky albedo. Page 9: ... the Rp3 clear-sky albedo output with MODIS CSD albedo for both broad-

**TCD**
band and narrowband albedo... Page 11: ...MODIS bands with diffuse radiation. For these bands, CSD albedo output of Rp3 is available. Note that the albedo determined for the seven MODIS bands are not used to compute a broadband albedo within Rp3. Exactly, the only major difference between clear-sky and cloudy-sky diffuse is associated with spectral shifts induced by clouds, which depends on the liquid water and ice water content of the cloud (See Van Dalum et al, 2019). As is stated in Van Dalum et al. (2019), variations in the clear-sky diffuse albedo in Rp3 are generally small, and only significantly change for high solar zenith angle, as the spectral distribution of energy is changed accordingly. Moreover, usually a large fraction of radiation is direct, reducing the impact of clear-sky diffuse radiation. We have added the following to the manuscript: Page 4: ...path for cloudy conditions. The difference between cloudy-diffuse and clear-sky diffuse albedo are thus only related to cloud and SZA induced spectral shifts in radiation. Furthermore, direct radiation dominates the clear-sky albedo signal except for very high SZA. As full radiation...

line 140: "Firstly, we assume that clean bubble-free ice has an albedo of approximately 0.6" - Pure, bubble-free ice technically has a much lower albedo than this, as described by the Fresnel equations. I assume this higher (measured?) albedo is caused by surface scattering and roughness. If so, this should be mentioned. We were wrong with stating that it was bubble-free ice that was measured. On the contrary, the blue ice has a high bubble concentration. We have changed the text accordingly (See comment (2) of the review by Stephen Warren).

line 154: What type of impurity is indicated with these concentrations? (Presumably soot). Page 5: The resulting soot concentration varies between...

line 169: "RACMO2 only allows for a fixed soot concentration." - But to be clear, the prescribed soot concentration varies over bare ice, and is only fixed over snow, correct? This distinction could use some clarification. Correct, we change the following:

Page 7: The only impurity type considered is soot, with a prescribed concentration of
5 ng g-1 for all snow layers. Although the concentration of soot in Greenland varies considerably over time and space, RACMO2 only allows for a fixed soot concentration in snow. If a layer is identified as bare ice, it is prescribed by the spatially variable soot concentration of Fig. 1b. For snow in the interior and when no melt occurs...

line 184-185: "But note that the MCD43A3 WSA product remains a slightly different albedo product than the clear-sky RACMO2 albedo it evaluates, which includes both direct and diffuse radiation." - As described earlier, however, both direct and diffuse clear-sky albedos are calculated by the model. Why not use the diffuse clear-sky albedo for comparison with MCD43 white-sky albedo. Wouldn't this be an apples-for-apples comparison? You are right that it would be ideal to compare diffuse clear-sky albedo with MODIS white-sky albedo. Unfortunately, even though it is calculated by the model, diffuse clear-sky albedo is not available as output in RACMO2, only total-sky and clear-sky (as in, direct and diffuse radiation combined) albedo is available. Therefore, we are limited to the comparison of clear-sky RACMO2 albedo with MODIS white sky albedo. We clarify this a bit more: Page 7: While RACMO2 calculates the direct and diffuse albedo, it only produces total-sky and clear-sky albedo output, which includes both direct and diffuse radiation. Therefore, we have to note that the MCD43A3 CSD albedo product remains a slightly different albedo product than the clear-sky RACMO2 albedo it evaluates.

line 199: Again, I believe Stroeve et al (2013) use version of 5 of this product. See the changes under (1) of this review.

line 357: "... high SZA... The spectral albedo of IR radiation is low, hence the broadband albedo drops" - But the SZA grazing effect outweighs the spectral shift, producing \*higher\* albedo at higher SZA, doesn't it? Your language on p.2 suggests so: "... this increase of spectral albedo at large SZA is largely mitigated by the red shift..." (i.e., largely, but not entirely, mitigated). You are right that the albedo increases for high SZA, which compensates for the spectral shift. You are also right that we did not state correctly what we meant to say. We wanted to say that the albedo difference for Rp3
with Rp2 becomes increasingly bigger for high SZA, as the spectral shift effect is not properly captured in Rp2. The albedo increase with SZA does happen for both models (See Figure 3, which shows the broadband albedo as a function of SZA, with the red line the albedo scheme of Rp2, the solid black line of Rp3), but the albedo increase goes not as fast in Rp3 as in Rp2 due to the aforementioned spectral shifts.

[revised manuscript text omitted]

Please also note the supplement to this comment: https://tc.copernicus.org/preprints/tc-2020-118/tc-2020-118-AC1-supplement.pdf

**TCD**